# Inverse Bridge Matching Distillation

**Nikita Gushchin** [1 2]  **David Li** [* 1 3]  **Daniil Selikhanovych** [* 1 4 5]  **Evgeny Burnaev** [1 2]  **Dmitry Baranchuk** [4]
**Alexander Korotin** [1 2]

## Abstract

Learning diffusion bridge models is easy; making them fast and practical is an art. Diffusion bridge models (DBMs) are a promising extension of diffusion models for applications in image-to-image translation. However, like many modern diffusion and flow models, DBMs suffer from the problem of slow inference. To address it, we propose a novel distillation technique based on the inverse bridge matching formulation and derive the tractable objective to solve it in practice. Unlike previously developed DBM distillation techniques, the proposed method can distill both conditional and unconditional types of DBMs, distill models in a one-step generator, and use only the corrupted images for training. We evaluate our approach for both conditional and unconditional types of bridge matching on a wide set of setups, including super-resolution, JPEG restoration, sketch-to-image, and other tasks, and show that our distillation technique allows us to accelerate the inference of DBMs from 4x to 100x and even provide better generation quality than used teacher model depending on particular setup. We provide the code at https://github.com/ngushchin/IBMD.

## 1. Introduction

Diffusion Bridge Models (DBMs) represent a specialized class of diffusion models designed for data-to-data tasks, such as image-to-image translation. Unlike standard diffusion models, which operate by mapping noise to data (Ho et al., 2020; Sohl-Dickstein et al., 2015), DBMs construct diffusion processes directly between two data distributions (Peluchetti, 2023a; Liu et al., 2022b; Somnath et al., 2023;

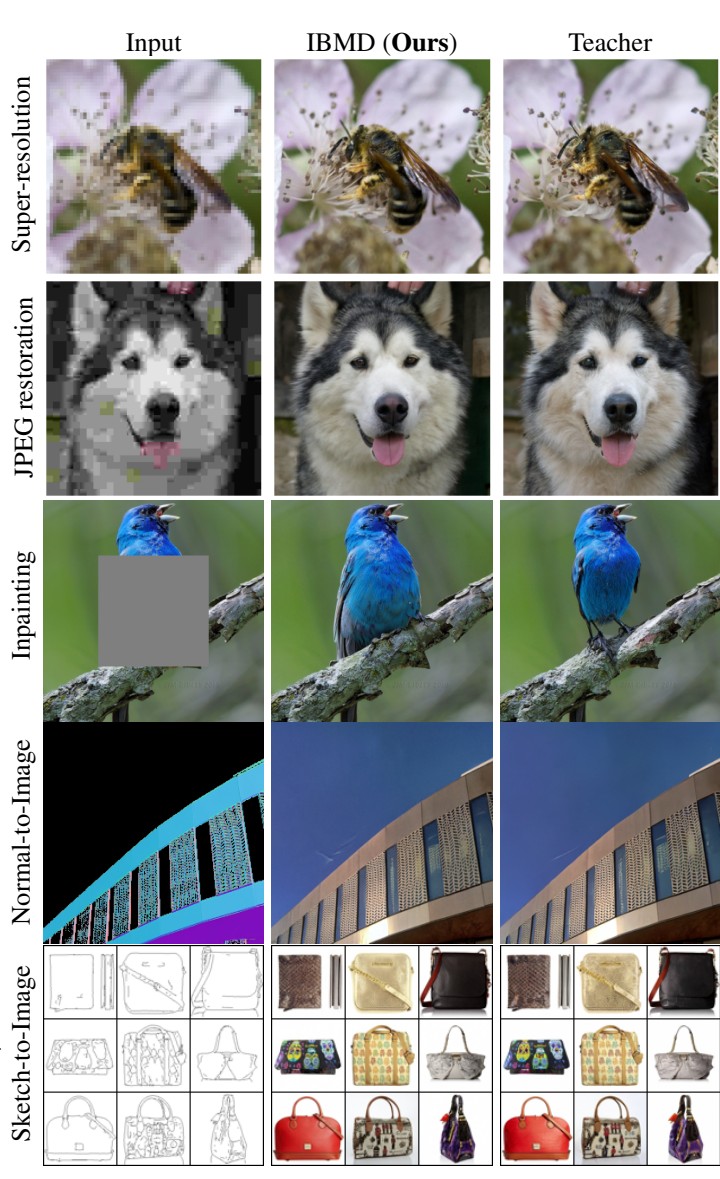

*Figure 1.* Outputs of DBMs models distilled by our **Inverse Bridge Matching Distillation (IBMD)** approach on various image-to-image translation tasks and datasets (§5). Teachers use NFE$\geq 500$ steps, while IBMD distilled models use NFE$\leq 4$.

Zhou et al., 2024a; Yue et al., 2024; Shi et al., 2023; De Bortoli et al., 2023). This approach allows DBMs to modify only the necessary components of the data, starting from an input sample rather than generating it entirely from Gaus-

---

[*]Equal contribution [1]Skolkovo Institute of Science and Technology [2]Artificial Intelligence Research Institute [3]Moscow Institute of Physics and Technology [4]Yandex Research [5]HSE University. Correspondence to: Nikita Gushchin <n.gushchin@skoltech.ru>, Alexander Korotin <a.korotin@skoltech.ru>.

*Proceedings of the 42$^{nd}$ International Conference on Machine Learning*, Vancouver, Canada. PMLR 267, 2025. Copyright 2025 by the author(s).

sian noise. As a result, DBMs have demonstrated impressive performance in image-to-image translation problems.

The rapid development of DBMs has led to two dominant approaches, usually considered separately. The first branch of approaches (Peluchetti, 2023a; Liu et al., 2022b; 2023a; Shi et al., 2023; Somnath et al., 2023) considered the construction of diffusion between two arbitrary data distributions performing **Unconditional Bridge Matching** (also called the Markovian projection) of a process given by a mixture of diffusion bridges. The application of this branch includes different data like images (Liu et al., 2023a; Li et al., 2023), audio (Kong et al., 2025) and biological tasks (Somnath et al., 2023; Tong et al., 2024) not only in paired but also in unpaired setups using its relation to the Schrödinger Bridge problem (Shi et al., 2023; Gushchin et al., 2024). The second direction follows a framework closer to classical diffusion models, using forward diffusion to gradually map to the point of different distibution rather than mapping distribution to distribution as in previous case (Zhou et al., 2024a; Yue et al., 2024). While these directions differ in theoretical formulation, their practical implementations are closely related; for instance, models based on forward diffusion can be seen as performing **Conditional Bridge Matching** with additional drift conditions (De Bortoli et al., 2023).

Similar to classical DMs, DBMs also exhibit multi-step sequential inference, limiting their adoption in practice. Despite the impressive quality shown by DBMs in the practical tasks, only a few approaches were developed for their acceleration, including more advanced sampling schemes (Zheng et al., 2024; Wang et al., 2024) and consistency distillation (He et al., 2024), adapted for bridge models. While these approaches significantly improve the efficiency of DBMs, some unsolved issues remain. The first one is that the developed distillation approaches are directly applicable only for DBMs based on the Conditional Bridge Matching, i.e., no universal distillation method can accelerate any DBMs. Also, due to some specific theoretical aspects of DBMs, consistency distillation cannot be used to obtain the single-step model (He et al., 2024, Section 3.4).

**Contributions.** To address the above-mentioned issues of DBMs acceleration, we propose a new distillation technique based on the inverse bridge matching problem, which has several advantages compared to existing methods:

1. **Universal Distillation.** Our distillation technique is applicable to DBMs trained with both conditional and unconditional regimes, making it the first distillation approach introduced for unconditional DBMs.

2. **Single-Step and Multi-step Distillation.** Our distillation is capable of distilling DBMs into generators with any specified number of steps, including the distillation of DBMs into one-step generators.

3. **Target data-free distillation.** Our method does not require the target data domain to perform distillation.

4. **Better quality of distilled models.** Our distillation technique is tested on a wide set of image-to-image problems for conditional and unconditional DBMs in both one and multi-step regimes. It demonstrates improvements compared to the previous acceleration approaches including DBIM (Zheng et al., 2024) and CDBM (He et al., 2024).

## 2. Background

In this paper, we propose a universal distillation framework for both conditional and unconditional DBMs. To not repeat fully analogical results for both cases, we denote by this color the additional conditioning on $x_T$ used for the conditional models, i.e. for the unconditional case this conditioning is not used.

### 2.1. Bridge Matching

We start by recalling the bridge matching method (Peluchetti, 2023b;a; Liu et al., 2022b; Shi et al., 2023). Consider two probability distributions $p(x_0)$ and $p(x_T)$ on $\mathbb{R}^D$ dimensional space, which represent target and source domains, respectively. For example, in an image inverse problem, $p(x_0)$ represents the distribution of *clean* images and $p(x_T)$ the distribution of *corrupted* images. Also consider a coupling $p(x_0, x_T)$ of these two distributions, which is a probability distribution on $\mathbb{R}^D \times \mathbb{R}^D$. Coupling $p(x_0, x_T)$ can be provided by paired data or constructed synthetically, i.e., just using the independent distribution $p(x_0, x_T) = p(x_0)p(x_T)$. Bridge Matching aims to construct the diffusion that transforms source distribution $p(x_T)$ to target distribution $p(x_0)$ based on given coupling $p(x_0, x_T)$ and specified *diffusion bridge*.

**Diffusion bridges.** Consider forward-time diffusion $Q$ called "Prior" on time horizon $[0, T]$ represented by the stochastic differential equation (SDE):

$$Prior\ Q: \quad dx_t = f(x_t, t)dt + g(t)dw_t, \quad (1)$$

$$f(x_t, t) : \mathbb{R}^D \times [0, T] \to \mathbb{R}^D, \quad g(t) : [0, T] \to \mathbb{R}^D,$$

where $f(x_t, t)$ is a drift function, $g(t)$ is the noise schedule function and $dw_t$ is the differential of the standard Wiener process. By $q(x_t|x_s)$, we denote the transition probability density of prior process $Q$ from time $s$ to time $t$. *Diffusion bridge* is a conditional process $Q_{|x_0, x_T}$, which is obtained by pinning down starting and ending points $x_0$ and $x_T$. This diffusion bridge can be derived from prior process $Q$ using the Doob-h transform (Doob & Doob, 1984):

$$Diffusion\ Bridge\ Q_{|x_0, x_T} : x_0, x_T\ are\ fixed, \quad (2)$$

$$dx_t = \{f(x_t, t)dt + g^2(t)\nabla_{x_t} \log q(x_T|x_t)\}dt + g(t)dw_t,$$

For this diffusion bridge we denote the distribution at time $t$ of the diffusion bridge $Q_{|x_0, x_T}$ by $q(x_t|x_0, x_T)$.

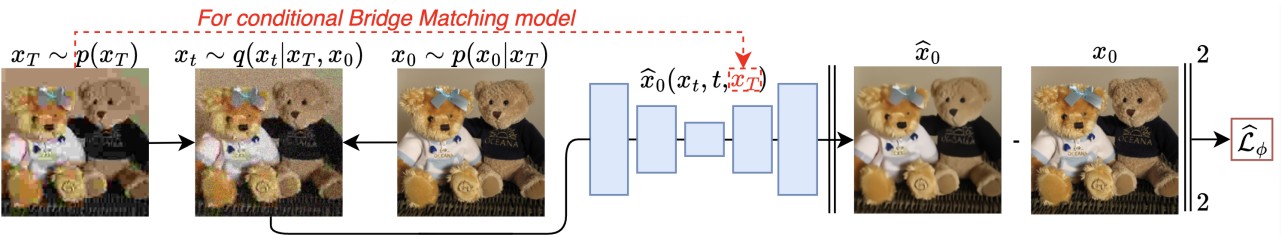

*Figure 2.* **Overview of (Conditional) Bridge Matching with $\widehat{x}_0$ reparameterization.** The process begins by sampling a pair $(x_0, x_T)$ from the data coupling $p(x_0, x_T)$. An intermediate sample $x_t$ is then drawn from the diffusion bridge $q(x_t|x_0, x_T)$ at a random time $t \sim U[0, T]$. The model $\widehat{x}_0$ is trained with an MSE loss to reconstruct $x_0$ from $x_t$. In the conditional setting (dashed red path), $\widehat{x}_0$ is also conditioned on $x_T$ as an additional input, leveraging information about the terminal state to improve reconstruction.

**Mixture of bridges.** Bridge Matching procedure starts with creating a *mixture of bridges* process $\Pi$. This process is represented as follows:

$$Mixture\ of\ Bridges\ \Pi:$$
$$\Pi(\cdot) = \int Q_{|x_0, x_T}(\cdot) p(x_0, x_T) dx_0 dx_T. \quad (3)$$

Practically speaking, the definition (3) means that to sample from a mixture of bridges $\Pi$, one first samples the pair $(x_0, x_T) \sim p(x_0, x_T)$ from data coupling and then samples trajectory from the bridge $Q_{|x_0, x_T}(\cdot)$.

**Bridge Matching problem.** The mixture of bridges $\Pi$ cannot be used for data-to-data translation since it requires first to sample a pair of data and then just inserts the trajectory. In turn, we are interested in constructing a diffusion, which can start from any sample $x_T \sim p(x_T)$ and gradually transform it to $x_0 \sim p(x_0)$. This can be done by solving the Bridge Matching problem (Shi et al., 2023, Proposition 2)

$$Bridge\ Matching\ problem: \quad (4)$$
$$BM(\Pi) \stackrel{\text{def}}{=} \underset{M \in \mathcal{M}}{\arg\min} \, KL(\Pi||M),$$

where $\mathcal{M}$ is the set of Markovian processes associated with some SDE and $KL(\Pi||M)$ is the KL-divergence between a constructed mixture of bridges $\Pi$ and diffusion $M$. It is known that the solution of Bridge Matching is the reversed-time SDE (Shi et al., 2023, Proposition 9):

$$The\ SDE\ of\ Bridge\ Matching\ solution: \quad (5)$$
$$dx_t = \{f_t(x_t) - g^2(t)v^*(x_t, t)\}dt + g(t)d\bar{w}_t,$$
$$x_T \sim p_T(x_T),$$

where $\bar{w}$ is a standard Wiener process when time $t$ flows backward from $t = T$ to $t = 0$, and $dt$ is an infinitesimal negative timestep. The drift function $v^*$ is obtained solving the following problem (Shi et al., 2023; Liu et al., 2023a):

*Bridge Matching problem with a tractable objective:* (6)
$$\min_{\phi} \mathbb{E}_{x_0, t, x_t} \left[ \|v_\phi(x_t, t) - \nabla_{x_t} \log q(x_t|x_0)\|^2 \right],$$
$$(x_0, x_T) \sim p(x_0, x_T), \ t \sim U([0, T]), \ x_t \sim q(x_t|x_0, x_T).$$

Time moment $t$ here is sampled according to the uniform distribution on the interval $[0, T]$.

**Relation Between Flow and Bridge Matching.** The Flow Matching (Liu et al., 2023b; Lipman et al., 2023) can be seen as the limiting case $\sigma \to 0$ of the Bridge Matching for particular example see (Shi et al., 2023, Appendix A.1).

## 2.2. Augmented (Conditional) Bridge Matching and Denoising Diffusion Bridge Models (DDBM)

For a given coupling $p(x_0, x_T) = p(x_0|x_T)p(x_T)$, one can use an alternative approach to build a data-to-data diffusion. Consider a set of Bridge Matching problems indexed by $x_T$ between $p_0 = p(x_0|x_T)$ and $p(x_T) = \delta_{x_T}(x)$ (delta measure centered at $x_T$). This approach is called Augmented Bridge Matching (De Bortoli et al., 2023). The key difference of this version in practice is that it introduces the condition of the drift function $v^*(x_t, t, x_T)$ on the starting point $x_T$ in the reverse time diffusion (5):

$$dx_t = \{f_t(x_t) - g^2(t)v^*(x_t, t, x_T)\}dt + g(t)d\bar{w}_t.$$

The drift function $v^*$ can be recovered almost in the same way just by the addition of this condition on $x_T$:

*Augmented (Conditional) Bridge Matching Problem.*
$$\min_{\phi} \mathbb{E}_{x_0, t, x_t, x_T} \left[ \|v_\phi(x_t, t, x_T) - \nabla_{x_t} \log q(x_t|x_0)\|^2 \right],$$
$$(x_0, x_T) \sim p(x_0, x_T), \text{and } x_t \sim q(x_t|x_0, x_T).$$

Since the difference is the addition of conditioning on $x_T$, we call this approach *Conditional Bridge Matching*.

**Relation to DDBM.** As was shown in the Augmented Bridge Matching (De Bortoli et al., 2023), the conditional Bridge Matching is equivalent to the Denoising Diffusion Bridge Model (DDBM) proposed in (Zhou et al., 2024a). The difference is that in DDBM, the authors learn the score function of $s(x_t, x_T, t)$ conditioned on $x_T$ of a process for which $x_0 \sim p(x_0|x_T)$ and $q(x_t) \sim q(x_t|x_0, x_T)$: Then, it is combined with the drift of forward Doob-h transform (5) to get the reverse SDE drift $v(x_t, t, x_T)$:

$$v(x_t, t, x_T) = s(x_t, x_T, t) - \nabla_{x_t} \log q(x_T|x_t),$$

$$dx_t = \{f(x_t, t)dt - g^2(t)v(x_t, t, x_T)\}dt + g(t)d\bar{w}_t,$$

or reverse probability flow ODE drift:

$$v_{\text{ODE}}(x_t, t, x_T) = \frac{1}{2}s(x_t, x_T, t) - \nabla_{x_t}\log q(x_T|x_t),$$
$$dx_t = \{f(x_t, t)dt - g^2(t)v_{\text{ODE}}(x_t, t, x_T)\}dt,$$

which is used for consistency distillation in (He et al., 2024).

### 2.3. Practical aspects of Bridge Matching

**Priors used in practice.** In practice (He et al., 2024; Zhou et al., 2023; Zheng et al., 2024), the drift of the prior process is usually set to be $f(x_t, t) = f(t)x_t$, i.e, it depends linearly on $x_t$. For this process the transitional distribution $q(x_t|x_0) = \mathcal{N}(x_t|\alpha_t x_0, \sigma_t^2 I)$ is Gaussian, where:

$$f(t) = \frac{d\log\alpha_t}{dt}, \quad g^2(t) = \frac{d\sigma_t^2}{dt} - 2\frac{d\log\alpha_t}{dt}\sigma_t^2.$$

The bridge process distribution is also a Gaussian $q(x_t|x_0, x_T) = \mathcal{N}(x_T|a_t x_T + b_t x_0, c_t^2 I)$ with coefficients:

$$a_t = \frac{\alpha_t}{\alpha_T}\frac{\text{SNR}_T}{\text{SNR}_t}, b_t = \alpha_t\left(1 - \frac{\text{SNR}_T}{\text{SNR}_t}\right),$$
$$c_t^2 = \sigma_t^2\left(1 - \frac{\text{SNR}_T}{\text{SNR}_t}\right),$$

where $\text{SNR}_t = \frac{\alpha_t^2}{\sigma_t^2}$ is the signal-to-noise ratio at time t.

**Data prediction reparameterization.** The regression target of the loss function (6) for the priors with the drift $v(x_t, t)$ is given by $\nabla_{x_t}\log q(x_t|x_0) = -\frac{x_t - \alpha_t x_0}{\sigma_t^2}$. Hence, one can use the parametrization $v(x_t, t, x_T) = -\frac{x_t - \alpha_t \hat{x}_0(x_t, t, x_T)}{\sigma_t^2}$ and solve the equivalent problem:

*Reparametrized (Conditional) Bridge Matching problem:*
$$\min_\phi \mathbb{E}_{x_0, t, x_t, x_T}\left[\lambda(t)\|\hat{x}_0^\phi(x_t, t, x_T) - x_0\|^2\right], \quad (7)$$

$$(x_0, x_T) \sim p(x_0, x_T), \ t \sim U([0, T]), \ x_t \sim q(x_t|x_0, x_T),$$

where $\lambda(t)$ is any positive weighting function. Note that $x_T$ is used only for the Conditional Bridge Matching model.

### 2.4. Difference Between Acceleration of Unconditional and Conditional DBMs

Since both conditional and unconditional approaches learn drifts of SDEs, they share the same problems of long inference. However, these models significantly differ in the approaches that can accelerate them. The source of this difference is that Conditional Bridge Matching considers the set of problems of reversing diffusion, which gradually transforms distribution $p(x_0|x_T)$ to the fixed point $x_T$. Furthermore, the forward diffusion has simple analytical drift

and Gaussian transitional kernels. Thanks to it, for each $x_T$ to sample, one can use the probability flow ODE and ODE-solvers or hybrid solvers to accelerate sampling (Zhou et al., 2024a) or use consistency distillation of bridge models (He et al., 2024). Another beneficial property is that one can consider a non-Markovian forward process to develop a more efficient sampling scheme proposed in DBIM (Zheng et al., 2024) similar to Denoising Diffusion Implicit Models (Song et al., 2021). However, in the Unconditional Bridge Matching problem, the forward diffusion process, which maps $p(x_0)$ to $p(x_T)$ without conditioning on specific point $x_T$, is unknown. Hence, the abovementioned methods cannot be used to accelerate this model type.

## 3. IBMD: Inverse Bridge Matching Distillation

This section describes our proposed universal approach to distill the both Unconditional and (Conditional) Bridge Matching models $v^*$ (called the teacher model) into a few-step generator *using only the corrupted data* $p_T(x_T)$. The key idea of our method is to consider the inverse problem of finding the mixture of bridges $\Pi_\theta$, for which Bridge Matching provides the solution $v_\theta$ with the same drift as the given teacher model $v^*$. We formulate this task as the optimization problem (§3.1). However, gradient methods cannot solve this optimization problem directly due to the absence of tractable gradient estimation. To avoid this problem, we prove a theorem that allows us to reformulate the inverse problem in the tractable objective for gradient optimization (§3.2). Then, we present the fully analogical results for the Conditional Bridge Matching case in (§3.3). Next, we present the multi-step version of distillation (§3.5) and the final algorithm (§3.4). We provide the proofs for all considered theorems and propositions in Appendix A.

### 3.1. Bridge Matching Distillation as Inverse Problem

In this section, we focus on the derivation of our distillation method for the case of Unconditional Bridge Matching. Consider the fitted teacher model $v^*(x_t, t)$, which is an SDE drift of some process $M^* = \text{BM}(\Pi^*)$, where $\Pi^*$ constructed using some data coupling $p^*(x_0, x_T) = p^*(x_0|x_T)p(x_T)$. We parametrize $p_\theta(x_0, x_T) = p_\theta(x_0|x_T)p(x_T)$ and aim to find such $\Pi_\theta$ build on $p_\theta(x_0, x_T)$, that $\text{BM}(\Pi^*) = \text{BM}(\Pi_\theta)$. In practice, we parametrize $p_\theta(x_0|x_T)$ by the stochastic generator $G_\theta(x_T, z), z \sim \mathcal{N}(0, I)$, which generates samples based on input $x_T \sim p(x_T)$ and the gaussian noise $z$. Now, we formulate the inverse problem as follows:

$$\min_\theta \text{KL}(\text{BM}(\Pi_\theta)||M^*). \quad (8)$$

Note, that since the objective (8) is the KL-divergence between $\text{BM}(\Pi_\theta)$ and $M^*$, it is equal to 0 if and only if $\text{BM}(\Pi_\theta)$ and $M^*$ coincide. Furthermore, using the disinte-

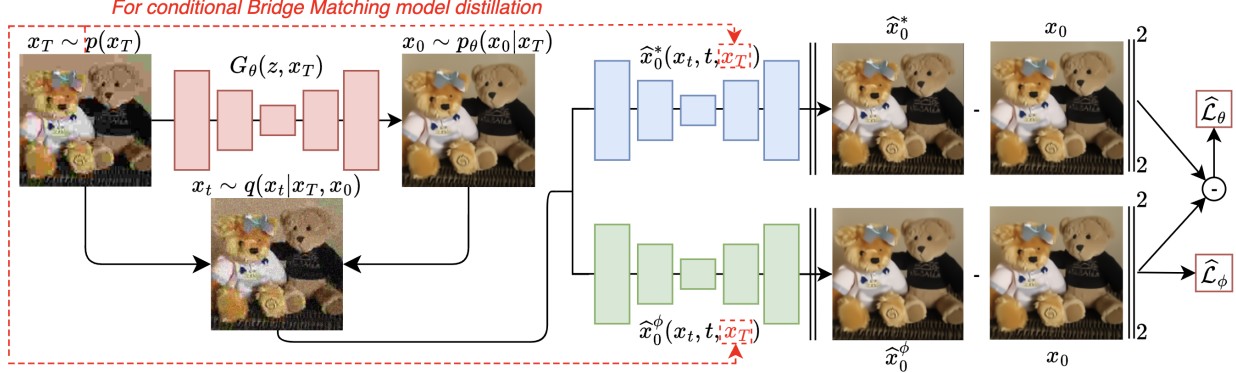

*Figure 3.* **Overview of our method Inverse Bridge Matching Distillation (IBMD).** The goal is to distill a trained (Conditional) Bridge Matching model into a generator $G_\theta(z, x_T)$, which learns to produce samples using the corrupted data $p(x_T)$. Generator $G_\theta(z, x_T)$ defines the coupling $p_\theta(x_0, x_T) = p_\theta(x_0|x_T)p(x_T)$ and we aim to learn the generator in such way that Bridge Matching with $p_\theta(x_0, x_T)$ produces the same (Conditional) Bridge Matching model $\widehat{x}_0^\phi = \widehat{x}_0^\theta$. To do so, we learn a bridge model $\widehat{x}_0^\phi$ using coupling $p_\theta$ in the same way as the teacher model was learned. Then, we use our novel objective given in Theorem 3.2 to update the generator model $G_\theta$.

gration and Girsanov theorem (Vargas et al., 2021; Pavon & Wakolbinger, 1991), we have the following result:

**Proposition 3.1** (Inverse Bridge Matching problem)**.** *The inverse problem* (8) *is equivalent to*

$$\min_\theta \mathbb{E}_{x_t, t}\big[\lambda(t)||v(x_t, t) - v^*(x_t, t)||^2\big], \quad s.t. \quad (9)$$

$$v = \arg\min_{v'} \mathbb{E}_{x_t, t, x_0}\big[||v'(x_t, t) - \nabla_{x_t} \log q(x_t|x_0)||^2\big],$$

$$(x_0, x_T) \sim p_\theta(x_0, x_T), \ t \sim U([0, T]), \ x_t \sim q(x_t|x_0, x_T),$$

*where $\lambda(t)$ is any positive weighting function.*

Thus, this is the **constrained** problem, where the drift $v$ is the result of Bridge Matching for coupling $p_\theta(x_0, x_T)$ parametrized by the generator $G_\theta$. Unfortunately, there is no clear way to use this objective efficiently for optimizing a generator $G_\theta$ since it would require gradient backpropagation through the argmin of the Bridge Matching problem.

### 3.2. Tractable objective for the inverse problem

In this section, we introduce our new **unconstrained** reformulation for the inverse problem (9), which admits direct optimization using gradient methods:

**Theorem 3.2** (Tractable inverse problem reformulation)**.** *The constrained inverse problem* (9) *w.r.t $\theta$ is equivalent to the unconstrained optimization problem:*

$$\min_\theta \Big[\mathbb{E}_{x_t, t, x_0}\big[\lambda(t)||v^*(x_t, t) - \nabla_{x_t} \log q(x_t|x_0)||^2\big] -$$

$$\min_\phi \mathbb{E}_{x_t, t, x_0}\big[\lambda(t)||v_\phi(x_t, t) - \nabla_{x_t} \log q(x_t|x_0)||^2\big]\Big],$$

$$(x_0, x_T) \sim p_\theta(x_0, x_T), \ t \sim U([0, T]), \ x_t \sim q(x_t|x_0, x_T),$$

*Where the constraint in the original inverse problem* (9) *is relaxed by introducing the inner bridge matching problem.*

This is the general result that can applied with any diffusion bridge. For the priors with with drift $f(x_t, t) = f(t)x_t$, we present its reparameterized version.

**Proposition 3.3** (Reparameterized tractable inverse problem)**.** *Using the reparameterization (§2.3) for the prior with the linear drift $f(x_t, t) = f(t)x_t$, the inverse problem in Theorem 3.2 is equivalent to:*

$$\min_\theta \Big[\mathbb{E}_{x_t, t, x_0}\big[\lambda(t)||\widehat{x}_0^*(x_t, t) - x_0||^2\big] -$$

$$\min_\phi \mathbb{E}_{x_t, t, x_0}\big[\lambda(t)||\widehat{x}_0^\phi(x_t, t) - x_0||^2\big]\Big],$$

$$(x_0, x_T) \sim p_\theta(x_0, x_T), \ t \sim U([0, T]), \ x_t \sim q(x_t|x_0, x_T).$$

Thanks to the unconstrained reformulation, this problem admits explicit gradients with respect to the generator $G_\theta$, as all samples $(x_0, x_T, x_t)$ are obtained via reparameterizable transformations: $x_0 = G_\theta(x_T, z)$ with $z \sim \mathcal{N}(0, I)$, and $x_t \sim q(x_t \mid x_0, x_T)$, where $q(x_t \mid x_0, x_T)$ is a Gaussian distribution (under priors with linear drift $f(x_t, t) = f(t)x_t$). This enables differentiability of the entire objective, which involves an expectation over $p_\theta(x_0, x_T)$, and allows optimization using standard gradient-based methods.

**Interpretation of the auxiliary model $\phi$.** Note that the minimal value of the inner problem is the averaged variance of $x_0 \sim p_\theta(x_0 \mid x_t, x_T)$:

$$\min_\phi \mathbb{E}_{x_t, t, x_0}\left[\lambda(t)\left\|\widehat{x}_0^\phi(x_t, t) - x_0\right\|^2\right] =$$

$$\mathbb{E}_{x_t, t, x_0}\left[\lambda(t)\left\|\mathbb{E}_{p_\theta(x_0|x_t)}[x_0] - x_0\right\|^2\right] =$$

$$\mathbb{E}_{x_t, t}\Big(\lambda(t)\underbrace{\Big[\mathbb{E}_{p_\theta(x_0|x_t)}\Big[\big\|\mathbb{E}_{p_\theta(x_0|x_t)}[x_0] - x_0\big\|^2\Big]\Big]}_{\text{Variance of } p_\theta(x_0|x_t)}\Big).$$

For $t = T$, this is directly the variance of the generator $x_0 \sim p_\theta(x_0 \mid x_T)$. Since this part comes with a negative

sign in the objective, its minimization enforces the generator to produce more diverse outputs and avoid collapsing.

### 3.3. Distillation of conditional Bridge Matching models

Since Conditional Bridge Matching is, in essence, a set of Unconditional Bridge Matching problems for each $x_T$ (§2.2), the analogical results hold just by adding the conditioning on $x_T$ for $v$, i.e., using $v(x_t, t, x_T)$ or $\widehat{x}_0$, i.e. using $\widehat{x}_0(x_t, t, x_T)$. Here, we provide the final reparametrized formulation, which we use in our experiments:

**Theorem 3.4** (Reparameterized tractable inverse problem for conditional bridge matching)**.**

$$\min_{\theta}\Big[\mathbb{E}_{x_t,t,x_0,x_T}\big[\lambda(t)\|\widehat{x}_0^*(x_t,t,x_T)-x_0\|^2\big]- \quad (10)$$
$$\min_{\phi}\mathbb{E}_{x_t,t,x_0,x_T}\big[\lambda(t)\|\widehat{x}_0^\phi(x_t,t,x_T)-x_0\|^2\big]\Big],$$
$$(x_0,x_T)\sim p_\theta(x_0,x_T),\ t\sim U([0,T]),\ x_t\sim q(x_t|x_0,x_T).$$

*where $\lambda(t)$ is some positive weight function.*

To use it in practice, we parameterize $\widehat{x}_0(x_t, t, x_T)$ by a neural network with an additional condition on $x_T$.

### 3.4. Algorithm

We provide a one-step Algorithm 1 that solves the inverse Bridge Matching problem in the reformulated version that we use in our experiments. We provide a visual abstract of it in Figure 3. Note that a teacher in the velocity parameterization $v^*(x_t, t)$ can be easily reparameterized (§2.3) in $x_0$-prediction model using $\widehat{x}^*(x_t, t) = \frac{\sigma_t^2 v^*(x_t,t)+x_t}{\alpha_t}$.

### 3.5. Mulitistep distillation

We also present a multi-step modification of our distillation technique if a one-step generator struggles to distill the models, e.g., in inpainting setups, where the corrupted image $x_T$ contains less information. Our multi-step technique is inspired by similar approaches used in diffusion distillation methods (Yin et al., 2024a, DMD) and aims to avoid training/inference distribution mismatch.

We choose $N$ timesteps $\{0 < t_1 < t_2 < ... < t_N = T\}$ and add additional time input to our generator $G_\theta(x_t, z, t)$. For the conditional Bridge Matching case, we also add conditions on $x_T$ and use $G_\theta(x_t, z, t, x_T)$. To perform inference, we alternate between getting prediction from the generator $\widetilde{x}_0 = G_\theta(x_t, z, t)$ and using posterior sampling $q(x_{t_{n-1}}|\widetilde{x}_0, x_{t_n})$ given by the diffusion bridge. To train the generator in the multi-step regime, we use the same procedure as in one step except that to get input $x_t$ for intermediate times $t_n < t_N$, we first perform inference of our generator to get $x_0$ and then use bridge $q(x_t|\widetilde{x}_0, x_T)$.

## 4. Related work

**Diffusion Bridge Models (DBMs) acceleration.** Unlike a wide scope of acceleration methods developed for classical diffusion/flow models, only a few approaches were developed for DBM acceleration. Acceleration methods include more advanced samplers (Zheng et al., 2024; Wang et al., 2024) based on a reformulated forward diffusion process as a non-markovian process inspired by Denoising Diffusion Implicit Models (Song et al., 2021). Also, there is a distillation method based on the distilling probability-flow ODE into a few steps using consistency models (He et al., 2024), which is applicable only for conditional DBMs. However, for theoretical reasons (He et al., 2024, Section 3.4), consistency models for Diffusion Bridges cannot be distilled into one-step generators. Unlike existing distillation methods, our method is applicable to both conditional and unconditional DBMs and can distill into a one-step generator.

**Related diffusion and flow models distillation techniques.** Among the methods developed for the distillation of classical diffusion and flow models, the most related to our work are methods based on simultaneous training of few-step generators and auxiliary "fake" model, that predict score or drift function for the generator (Yin et al., 2024b;a; Zhou et al., 2024b; Huang et al., 2024). Unlike these approaches, we consider the distillation of Diffusion Bridge Models - the generalization of flow and diffusion models.

Furthermore, previous distillation methods for diffusion and flow models rely on marginal-based losses such as Fisher divergence, these approaches do not account for the full structure of path measures. This limitation becomes critical in the context of Diffusion Bridge Models (DBMs), where the dynamic aspects of the forward and reverse processes play a fundamental role. To better motivate the need for our KL-based objective, we next discuss the conceptual differences between KL divergence of path measures and Fisher divergence, illustrating why Fisher-based objectives like those used in SiD (Zhou et al., 2024b) are insufficient in the general setting of bridge matching. Consider two reverse-time diffusions $D_1$ and $D_2$ given by the same starting distribution $p(x_T)$ and SDEs:

$$D_1 : dx_t = v(x_t,t)dt + g^2(t)d\bar{w}_t, \quad x_T \sim p(x_T),$$
$$D_2 : dx_t = \widehat{v}(x_t,t)dt + g^2(t)d\bar{w}_t, \quad x_T \sim p(x_T)$$

Let $p(x_t)$ and $\widehat{p}(x_t)$ be the corresponding marginals. Then the KL divergence and Fisher divergence are given by:

$$\mathrm{KL}(D_1||D_2) = \mathbb{E}_{t,p_t(x_t)}\Big[\frac{1}{2g^2(t)}\|v(x_t,t)-\widehat{v}(x_t,t)\|^2\Big] +$$
$$\underbrace{\mathrm{KL}(p(x_T)||\widehat{p}(x_T))}_{=0 \text{ if } p(x_T)=\widehat{p}(x_T)}, \quad (11)$$

$$\mathrm{D}_{\mathrm{Fisher}}(D_1||D_2) = \mathbb{E}_{t,p(x_t)}\|\nabla_{x_t}\log p(x_t) - \nabla_{x_t}\log\widehat{p}(x_t)\|^2.$$

In SiD (Zhou et al., 2024b), Fisher divergence ($D_{\text{Fisher}}$) is averaged over time and compares only marginal distributions $p(x_t)$ and $\widehat{p}(x_t)$ of path measures. However, path measures with the same marginal distributions might not be equal; thus, in general, minimizing Fisher divergence does not guarantee that $D_1 \approx D_2$ as stochastic processes. For classical diffusion models, the forward drift $f(x_t, t)$ is fixed, and reverse drifts are fully determined by score functions:

$$\widehat{v}(x_t, t) = f(x_t, t) - g^2(t)\nabla_{x_t} \log \widehat{p}(x_t),$$
$$v(x_t, t) = f(x_t, t) - g^2(t)\nabla_{x_t} \log p(x_t).$$

Substituting these into the KL expression (11) shows that in this specific setting — with a fixed forward SDE — KL divergence between path measures becomes equivalent (up to a constant) to the time-averaged Fisher divergence between the marginals. This explains why Fisher-based methods like SiD (Zhou et al., 2024b) may succeed in this context.

However, this equivalence breaks down in the case of unconditional bridge matching. Here, the forward drift $f(x_t, t)$ is not fixed and depends on the data coupling $p(x_0, x_T)$. In turn, the forward drift $f_\theta(x_t, t)$ for the generated coupling $p_\theta(x_0, x_T)$ also depends on $\theta$. As a result, $f(x_t, t) \neq f_\theta(x_t, t)$, and the reverse drifts cannot be expressed solely in terms of marginal scores. Hence, KL divergence in the case of unconditional bridge matching is not equivalent to Fisher divergence between marginals. This difference is expected since, in the case of an unconditional diffusion bridge, one does not have a fixed forward process, which specifies the "dynamic part" of the measure. This highlights the importance of using KL divergence between path measures as a high-level objective instead of the previously used Fisher Divergence.

## 5. Experiments

This section highlights the applicability of our IBMD distillation method in both *unconditional* and *conditional* settings. To demonstrate this, we conducted experiments utilizing pretrained *unconditional* models used in $I^2$SB paper (Liu et al., 2023a). Then we evaluated IBMD in *conditional* settings using DDBM (Zhou et al., 2024a) setup (§5.2). For clarity, we denote our models as **IBMD-DDBM** and **IBMD-$I^2$SB**, indicating that the teacher model is derived from DDBM or $I^2$SB framework, respectively. We provide all the technical details in Appendix B.

### 5.1. Distillation of I2SB (5 setups)

Since known distillation and acceleration techniques are designed for the conditional models, there is no clear baseline for comparison. Thus, this section aims to demonstrate that our distillation technique significantly decreases NFE required to obtain the same quality of generation.

**Experimental Setup.** To test our approach for unconditional models, we consider models trained and published in

---

**Algorithm 1** Inverse Bridge Matching Distillation (IBMD)

**Input** : Teacher network $\widehat{x}_0^* : \mathbb{R}^D \times [0, T] \times \mathbb{R}^D \to \mathbb{R}^D$;
Bridge $q(x_t | x_0, x_T)$ used for training $x^*$;
Generator network $G_\theta : \mathbb{R}^D \times \mathbb{R}^D \to \mathbb{R}^D$;
Bridge network $\widehat{x}_0^\phi : \mathbb{R}^D \times [0, T] \times \mathbb{R}^D \to \mathbb{R}^D$;
Input distribution $p(x_T)$ accessible by samples;
Weights function $\lambda(t) : [0, T] \to \mathbb{R}^+$;
Batch size $N$; Number of student iterations $K$;
Number of bridge iterations $L$.

**Output** : Learned generator $G_\theta$ of coupling $p_\theta(x_0, x_T)$ for which Bridge Matching outputs drift $v \approx v^*$.

// Conditioning on $x_T$ is used only for distillation of Conditional Bridge Matching models.

**for** $k = 1$ **to** $K$ **do**
  **for** $l = 1$ **to** $L$ **do**
    Sample batch $x_T \sim p(x_T)$
    Sample batch of noise $z \sim \mathcal{N}(0, I)$
    $x_0 \leftarrow G_\theta(x_T, z)$
    Sample time batch $t \sim U[0, T]$
    Sample batch $x_t \sim q(x_t | x_0, x_T)$
    $\widehat{\mathcal{L}}_\phi \leftarrow \frac{1}{N} \sum_{n=1}^{N} \left[ \lambda(t) || \widehat{x}_0^\phi(x_t, t, x_T) - x_0 ||^2 \right]_n$
    Update $\phi$ by using $\frac{\partial \widehat{\mathcal{L}}_\phi}{\partial \phi}$
  Sample batch $x_T \sim p(x_T)$
  Sample batch of noise $z \sim \mathcal{N}(0, I)$
  $x_0 \leftarrow G_\theta(x_T, z)$
  Sample time batch $t \sim U[0, T]$
  Sample batch $x_t \sim q(x_t | x_0, x_T)$
  $\widehat{\mathcal{L}}_\theta \leftarrow \frac{1}{N} \sum_{n=1}^{N} \big[ \lambda(t) || \widehat{x}_0^*(x_t, t, x_T) - x_0 ||^2 - \lambda(t) || \widehat{x}_0^\phi(x_t, t, x_T) - x_0 ||^2 \big]_n$
  Update $\theta$ by using $\frac{\partial \widehat{\mathcal{L}}_\theta}{\partial \theta}$

---

$I^2$SB paper (Liu et al., 2023a), specifically (a) two models for the 4x super-resolution with bicubic and pool kernels, (b) two models for JPEG restoration using quality factor QF= 5 and QF= 10, and (c) a model for center-inpainting with a center mask of size $128 \times 128$ all of which were trained on ImageNet $256 \times 256$ dataset (Deng et al., 2009).

For all the setups we use the same train part of ImageNet dataset, which was used to train the used models. For the evaluation we follow the same protocol used in the $I^2$SB paper, i.e. use the full validation subset of ImageNet for super-resolution task and the $10'000$ subset of validation for other tasks. We report the same FID (Heusel et al., 2017) and Classifier Accuracy (CA) using pre-trained ResNet50 model metrics used in the $I^2$SB paper. We present our results in Table 1, Table 3, Table 2, Table 4 and Table 6. We provide the uncurated samples for all setups in Appendix C.

**Results.** For both super-resolution tasks (see Table 1, Table 3), our 1-step distilled model outperformed teacher model inference using all 1000 steps used in the training.

*Table 1.* Results on the image super-resolution task. Baseline results are taken from I²SB (Liu et al., 2023a).

| $4\times$ **super-resolution (bicubic)** | ImageNet ($256 \times 256$) | | |
|---|---|---|---|
| | NFE | FID ↓ | CA ↑ |
| DDRM (Kawar et al., 2022) | 20 | 21.3 | 63.2 |
| DDNM (Wang et al., 2023) | 100 | 13.6 | 65.5 |
| ΠGDM (Song et al., 2023) | 100 | 3.6 | 72.1 |
| ADM (Dhariwal & Nichol, 2021) | 1000 | 14.8 | 66.7 |
| CDSB (Shi et al., 2022) | 50 | 13.6 | 61.0 |
| I²SB (Liu et al., 2023a) | 1000 | 2.8 | 70.7 |
| IBMD-I²SB (**Ours**) | 1 | **2.6** | 72.3 |

*Table 3.* Results on the image super-resolution task. Baseline results are taken from I²SB (Liu et al., 2023a).

| $4\times$ **super-resolution (pool)** | ImageNet ($256 \times 256$) | | |
|---|---|---|---|
| | NFE | FID ↓ | CA ↑ |
| DDRM (Kawar et al., 2022) | 20 | 14.8 | 64.6 |
| DDNM (Wang et al., 2023) | 100 | 9.9 | 67.1 |
| ΠGDM (Song et al., 2023) | 100 | 3.8 | 72.3 |
| ADM (Dhariwal & Nichol, 2021) | 1000 | 3.1 | 73.4 |
| CDSB (Shi et al., 2022) | 50 | 13.0 | 61.3 |
| I²SB (Liu et al., 2023a) | 1000 | 2.7 | 71.0 |
| IBMD-I²SB (**Ours**) | 1 | **2.5** | 72.5 |

*Table 2.* Results on the image JPEG restoration task with QF=5. Baseline results are taken from I²SB (Liu et al., 2023a).

| **JPEG restoration, QF**$= 5$. | ImageNet ($256 \times 256$) | | |
|---|---|---|---|
| | NFE | FID ↓ | CA ↑ |
| DDRM (Kawar et al., 2022) | 20 | 28.2 | 53.9 |
| ΠGDM (Song et al., 2023) | 100 | 8.6 | 64.1 |
| Palette (Saharia et al., 2022) | 1000 | 8.3 | 64.2 |
| CDSB (Shi et al., 2022) | 50 | 38.7 | 45.7 |
| I²SB (Liu et al., 2023a) | 1000 | **4.6** | 67.9 |
| I²SB (Liu et al., 2023a) | 100 | 5.4 | 67.5 |
| IBMD-I²SB (**Ours**) | 1 | 5.2 | 66.6 |

*Table 4.* Results on the image JPEG restoration task with QF=10. Baseline results are taken from I²SB (Liu et al., 2023a).

| **JPEG restoration, QF**$= 10$. | ImageNet ($256 \times 256$) | | |
|---|---|---|---|
| | NFE | FID ↓ | CA ↑ |
| DDRM (Kawar et al., 2022) | 20 | 16.7 | 64.7 |
| ΠGDM (Song et al., 2023) | 100 | 6.0 | 71.0 |
| Palette (Saharia et al., 2022) | 1000 | 5.4 | 70.7 |
| CDSB (Shi et al., 2022) | 50 | 18.6 | 60.0 |
| I²SB (Liu et al., 2023a) | 1000 | **3.6** | 72.1 |
| I²SB (Liu et al., 2023a) | 100 | 4.4 | 71.6 |
| IBMD-I²SB (**Ours**) | 1 | 3.7 | 72.4 |

Note that our model does not use the clean training target data $p(x_0)$, only the corrupted $p(x_T)$, hence this improvement is not due to additional training using paired data. We hypothesize that it is because the teacher model introduces approximation error during many steps of sampling, which may accumulate. For both JPEG restoration (see Table 2, Table 4), our 1-step distilled generator provides the quality of generation close to the teacher model and achieves around 100x time acceleration. For the inpainting problem (see Table 6), we present the results for $1, 2$ and $4$ steps distilled generator. Our 2 and 4-step generators provide a quality similar to the teacher I²SB model, however, there is still some gap for the 1-step model. These models provide around 5x time acceleration. We hypothesize that this setup is harder since it requires to generate the entire center fragment from scratch, while in other tasks, there is already some good approximation given by corrupted images.

### 5.2. Distillation of DDBM (3 setups)

This section addresses two primary objectives: (1) demonstrating the feasibility of conditional model distillation within our framework and (2) comparing with the CDBM (He et al., 2024) - a leading approach in Conditional Bridge Matching distillation, presented into different models: CBD (consistency distillation) and CBT (consistency training).

**Experimental Setup.** For evaluation, we use the same setups used in competing methods (He et al., 2024; Zheng

et al., 2024). For the image-to-image translation task, we utilize the Edges→Handbags dataset (Isola et al., 2017) with a resolution of $64 \times 64$ pixels and the DIODE-Outdoor dataset (Vasiljevic et al., 2019) with a resolution of $256\times256$ pixels. For these tasks, we report FID and Inception Scores (IS) (Barratt & Sharma, 2018). For the image inpainting task, we use the same setup of center-inpainting as before.

**Results.** We utilized the same teacher model checkpoints and as in CDBM. We present the quantitative and qualitative results of IBMD on the image-to-image translation task in Table 5 and in Figures 12, 10 respectively. The competing methods, DBIM (Zhou et al., 2024a, Section 4.1) and CDBM (He et al., 2024, Section 3.4), cannot use single-step inference due to the singularity at the starting point $x_T$.

We trained our IBMD with 1 and 2 NFEs on the Edges→Handbags dataset. We surpass CDBM at 2 NFE, outperform the teacher at 100 NFE, and achieve performance comparable to the teacher at 50 NFE with 1 NFE, resulting in a $50\times$ acceleration. For the DIODE-Outdoor setup, we trained IBMD with 1 and 2 NFEs. We surpassed CBD in FID at 2 NFE, achieving results comparable to CBT with a slight drop in performance and maintaining strong performance at 1 NFE with minor quality reductions.

For image inpainting, we show in Table 6 quantitative results and in Figure 9 the quantitative results. We train IBMD with 4 NFE in this setup. It outperforms CBD and CBT at 4

*Table 5.* Results on the Image-to-Image Translation Task (Training Sets). Methods are grouped by NFE ($> 2, 2, 1$), with the best metrics bolded in each group. Baselines results are taken from CDBM.

| | NFE | Edges → Handbags ($64 \times 64$) | | DIODE-Outdoor ($256 \times 256$) | |
|---|---|---|---|---|---|
| | | FID ↓ | IS ↑ | FID ↓ | IS ↑ |
| DDIB (Su et al., 2023) | $\geq 40$ | 186.84 | 2.04 | 242.3 | 4.22 |
| SDEdit (Meng et al., 2022) | $\geq 40$ | 26.5 | 3.58 | 31.14 | 5.70 |
| Rectified Flow (Liu et al., 2022a) | $\geq 40$ | 25.3 | 2.80 | 77.18 | 5.87 |
| I$^2$SB (Liu et al., 2023a) | $\geq 40$ | 7.43 | 3.40 | 9.34 | 5.77 |
| DBIM (Zheng et al., 2024) | 50 | 1.14 | **3.62** | 3.20 | **6.08** |
| DBIM (Zheng et al., 2024) | 100 | **0.89** | **3.62** | **2.57** | 6.06 |
| CBD (He et al., 2024) | | 1.30 | 3.62 | 3.66 | 6.02 |
| CBT (He et al., 2024) | 2 | 0.80 | 3.65 | **2.93** | **6.06** |
| IBMD-DDBM (**Ours**) | | **0.67** | **3.69** | 3.12 | 5.92 |
| Pix2Pix (Isola et al., 2017) | 1 | 74.8 | **4.24** | 82.4 | 4.22 |
| IBMD-DDBM (**Ours**) | | **1.26** | 3.66 | **4.07** | **5.89** |

*Table 6.* Results on the Image Inpainting Task. Methods are grouped by NFE ($> 4, 4, 2, 1$), with the best metrics bolded in each group. Baselines results are taken from CDBM.

| Inpainting, Center ($128 \times 128$) | ImageNet ($256 \times 256$) | | |
|---|---|---|---|
| | NFE | FID ↓ | CA ↑ |
| DDRM (Kawar et al., 2022) | 20 | 24.4 | 62.1 |
| ΠGDM (Song et al., 2023) | 100 | 7.3 | **72.6** |
| DDNM (Wang et al., 2023) | 100 | 15.1 | 55.9 |
| Palette (Saharia et al., 2022) | 1000 | 6.1 | 63.0 |
| I$^2$SB (Liu et al., 2023a) | 10 | 5.4 | 65.97 |
| DBIM (Zheng et al., 2024) | 50 | 3.92 | 72.4 |
| DBIM (Zheng et al., 2024) | 100 | **3.88** | **72.6** |
| CBD (He et al., 2024) | | 5.34 | 69.6 |
| CBT (He et al., 2024) | 4 | 4.77 | 70.3 |
| IBMD-I$^2$SB (**Ours**) | | 5.1 | 70.3 |
| IBMD-DDBM (**Ours**) | | **4.03** | **72.2** |
| CBD (He et al., 2024) | | 5.65 | 69.6 |
| CBT (He et al., 2024) | 2 | 5.34 | 69.8 |
| IBMD-I$^2$SB (**Ours**) | | 5.3 | 65.7 |
| IBMD-DDBM (**Ours**) | | **4.23** | **72.3** |
| IBMD-I$^2$SB (**Ours**) | 1 | 6.7 | 65.0 |
| IBMD-DDBM (**Ours**) | | **5.87** | **70.6** |

NFE with a significant gap, surpassing both at 2 NFE and maintaining strong performance at 1 NFE while achieving teacher-level results (with 50 NFE) with a $12.5\times$ speedup.

**Concerns regarding the evaluation protocol used in prior works.** For Edges-Handbags and DIODE-Outdoor setups, we follow the evaluation protocol originally introduced in DDBM (Zhou et al., 2024a) and later used in works on acceleration of DDBM (Zheng et al., 2024; He et al., 2024). For some reason, this protocol implies evaluation of the train set. Furthermore, test sets of these datasets consist of a tiny fraction of images (around several hundred), making the usage of standard metrics like FID challenging due to high statistical bias or variance of their estimation. Still, to assess the quality of the distilled model on the test sets, we provide the uncurated samples produced by our distill

model and teacher model on these sets in Figures 13 and 11 in Appendix C. We also provide the uncurated samples on the train part in Figures 12 and 10 to compare models' behavior on train and test sets. From these results, we see that the teacher model exhibits overfitting on both setups, e.g., it produces exactly the same images as corresponding reference images. In turn, on the test sets, teacher models work well for the handbag setups, while on the test set of DIODE images, it exhibits mode collapse and produces gray images. Nevertheless, our distilled model shows exactly the same behavior in both sets, i.e., our IBMD approach precisely distills the teacher model as expected.

## 6. Discussion

**Potential impact.** DBMs are used for data-to-data translation in different domains, including images, audio, and biological data. Our distillation technique provides a universal and efficient way to address the long inference of DBMs, making them more affordable in practice.

**Limitations.** Our method alternates between learning an additional bridge model and updating the student, which may be computationally expensive. Moreover, the student optimization requires backpropagation through the teacher, additional bridge, and the generator network, making it 3x time more memory expensive than training the teacher.

## Acknowledgements

The work was supported by the grant for research centers in the field of AI provided by the Ministry of Economic Development of the Russian Federation in accordance with the agreement 000000C313925P4F0002 and the agreement with Skoltech №139-10-2025-033.

## Impact Statement

This paper presents work whose goal is to advance the field of Machine Learning. There are many potential societal consequences of our work, none which we feel must be specifically highlighted here.

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

# A. Proofs

Since all our theorems, propositions and proofs for the inverse Bridge Matching problems which is formulated for the already trained teacher model using some diffusion bridge, we assume all corresponding assumptions used in Bridge Matching. Extensive overview of them can be found in (Shi et al., 2023, Appendix C).

*Proof of Proposition 3.1.* Since both $\text{BM}(\Pi_\theta)$ and $M^*$ given by reverse-time SDE and the same distribution $p_T(x_T)$ the KL-divergence expressed in the tractable form using the disintegration and Girsanov theorem (Vargas et al., 2021; Pavon & Wakolbinger, 1991):

$$\text{KL}(\text{BM}(\Pi_\theta)||M^*) = \mathbb{E}_{x_t,t}\big[g^2(t)||v(x_t,t) - v^*(x_t,t)||^2\big],$$
$$(x_0, x_T) \sim p_\theta(x_0, x_T), t \sim U([0,T]), x_t \sim q(x_t|x_0, x_T).$$

The expectation is taken over the marginal distribution $p(x_t)$ of $\Pi_\theta$ since it is the same as for $\text{BM}(\Pi_\theta)$ (Shi et al., 2023, Proposition 2). In turn, the drift $v(x_t,t)$ is the drift of Bridge Matching using $\Pi_\theta$, i.e. $\text{BM}(\Pi_\theta)$:

$$v = \arg\min_{v'} \mathbb{E}_{x_t,t,x_0}\big[||v'(x_t,t) - \nabla_{x_t}\log q(x_t|x_0)||^2\big],$$
$$(x_0, x_T) \sim p_\theta(x_0, x_T), t \sim U([0,T]), x_t \sim q(x_t|x_0, x_T).$$

Combining this, the inverse problem can be expressed in a more tractable form:

$$\min_\theta \mathbb{E}_{x_t,t}\big[g^2(t)||v(x_t,t) - v^*(x_t,t)||^2\big], \quad \text{s.t.} \tag{12}$$
$$v = \arg\min_{v'} \mathbb{E}_{x_t,t,x_0}\big[||v'(x_t,t) - \nabla_{x_t}\log q(x_t|x_0)||^2\big]dt,$$
$$(x_0, x_T) \sim p_\theta(x_0, x_T), \ t \sim U([0,T]), \ x_t \sim q(x_t|x_0, x_T).$$

We can add positive valued weighting function $\lambda(t)$ for the constraint:

$$v = \arg\min_{v'} \mathbb{E}_{x_t,t,x_0}\big[\lambda(t)||v'(x_t,t) - \nabla_{x_t}\log q(x_t|x_0)||^2\big]dt,$$

since it is the MSE regression and its solution is conditional expectation for any weights given by:

$$v(x_t,t) = \mathbb{E}_{x_0|x_t,t}\big[\nabla_{x_t}\log q(x_t|x_0)\big].$$

We can add positive valued weighting function $\lambda(t)$ for the main functional:

$$\mathbb{E}_{x_t,t}\big[\lambda(t)||v(x_t,t) - v^*(x_t,t)||^2\big],$$

since it does not change the optimum value (which is equal to 0) and optimal solution, which is the mixture of bridges with the same drift as the teacher model. $\square$

*Proof of Theorem 3.2.* Consider inverse bridge matching optimization problem:

$$\min_\theta \mathbb{E}_{x_t,t}\big[\lambda(t)||v(x_t,t) - v^*(x_t,t)||^2\big], \quad \text{s.t.} \tag{13}$$
$$v = \arg\min_{v'} \mathbb{E}_{x_t,t,x_0}\big[||v'(x_t,t) - \nabla_{x_t}\log q(x_t|x_0)||^2\big],$$
$$(x_0, x_T) \sim p_\theta(x_0, x_T), \ t \sim U([0,T]), \ x_t \sim q(x_t|x_0, x_T).$$

First, note that since $v = \arg\min_{v'} \mathbb{E}_{x_t,t,x_0}\big[||v'(x_t,t) - \nabla_{x_t}\log q(x_t|x_0)||^2\big]$, i.e. minimizer of MSE functional it is given by conditional expectation as:

$$v(x_t,t) = \mathbb{E}_{x_0|x_t,t}\big[\nabla_{x_t}\log q(x_t|x_0)|x_t,t\big]. \tag{14}$$

Then note that:

$$\min_{v'} \mathbb{E}_{x_t,t,x_0}\big[\lambda(t)||v'(x_t,t) - \nabla_{x_t}\log q(x_t|x_0)||^2\big] =$$

$$\mathbb{E}_{x_t,t,x_0}\big[\lambda(t)\|v(x_t,t) - \nabla_{x_t}\log q(x_t|x_0)\|^2\big] =$$

$$\underbrace{\mathbb{E}_{x_t,t,x_0}\big[\lambda(t)\|v(x_t,t)\|^2\big]}_{\mathbb{E}_{x_t,t}\big[\lambda(t)\|v(x_t,t)\|^2\big]} - 2\mathbb{E}_{x_t,t,x_0}\big[\lambda(t)\langle v(x_t,t), \nabla_{x_t}\log q(x_t|x_0)\rangle\big] + \mathbb{E}_{x_t,t,x_0}\big[\lambda(t)\|\nabla_{x_t}\log q(x_t|x_0)\|^2\big] =$$

$$\mathbb{E}_{x_t,t}\big[\lambda(t)\|v(x_t,t)\|^2\big] - 2\mathbb{E}_{x_t,t}\Big[\lambda(t)\Big\langle v(x_t,t), \underbrace{\mathbb{E}_{x_0|x_t,t}\big[\nabla_{x_t}\log q(x_t|x_0)\big]}_{=v(x_t,t)}\Big\rangle\Big] + \mathbb{E}_{x_t,t,x_0}\big[\lambda(t)\|\nabla_{x_t}\log q(x_t|x_0)\|^2\big] =$$

$$\mathbb{E}_{x_t,t}\big[\lambda(t)\|v(x_t,t)\|^2\big] - 2\mathbb{E}_{x_t,t}\big[\lambda(t)\|v(x_t,t)\|^2\big] + \mathbb{E}_{x_t,t,x_0}\big[\lambda(t)\|\nabla_{x_t}\log q(x_t|x_0)\|^2\big] =$$

$$-\mathbb{E}_{x_t,t}\big[\lambda(t)\|v(x_t,t)\|^2\big] + \mathbb{E}_{x_t,t,x_0}\big[\lambda(t)\|\nabla_{x_t}\log q(x_t|x_0)\|^2\big]. \tag{15}$$

Hence, we derive that

$$\mathbb{E}_{x_t,t}\big[\lambda(t)\|v(x_t,t)\|^2\big] = \mathbb{E}_{x_t,t,x_0}\big[\lambda(t)\|\nabla_{x_t}\log q(x_t|x_0)\|^2\big] - \min_{v'}\mathbb{E}_{x_t,t,x_0}\big[\lambda(t)\|v'(x_t,t) - \nabla_{x_t}\log q(x_t|x_0)\|^2\big].$$

Now we use it to reformulate the initial objective:

$$\mathbb{E}_{x_t,t}\big[\lambda(t)\|v(x_t,t) - v^*(x_t,t)\|^2\big] =$$

$$\mathbb{E}_{x_t,t}\big[\lambda(t)\|v(x_t,t)\|^2\big] - 2\mathbb{E}_{x_t,t}\big[\lambda(t)\langle v(x_t,t), v^*(x_t,t)\rangle\big] + \mathbb{E}_{x_t,t}\big[\lambda(t)\|v^*(x_t,t)\|^2\big] =$$

$$\underbrace{\mathbb{E}_{x_t,t,x_0}\big[\lambda(t)\|\nabla_{x_t}\log q(x_t|x_0)\|^2\big] - \min_{v'}\mathbb{E}_{x_t,t,x_0}\big[\lambda(t)\|v'(x_t,t) - \nabla_{x_t}\log q(x_t|x_0)\|^2\big]}_{=\mathbb{E}_{x_t,t}\big[\lambda(t)\|v(x_t,t)\|^2\big]} -$$

$$2\mathbb{E}_{x_t,t}\big[\lambda(t)\langle v(x_t,t), v^*(x_t,t)\rangle\big] + \mathbb{E}_{x_t,t}\big[\lambda(t)\|v^*(x_t,t)\|^2\big] =$$

$$\mathbb{E}_{x_t,t,x_0}\big[\lambda(t)\|\nabla_{x_t}\log q(x_t|x_0)\|^2\big] - 2\mathbb{E}_{x_t,t}\big[\lambda(t)\langle v(x_t,t), v^*(x_t,t)\rangle\big] + \underbrace{\mathbb{E}_{x_t,t}\big[\lambda(t)\|v^*(x_t,t)\|^2\big]}_{\mathbb{E}_{x_t,t,x_0}\big[\lambda(t)\|v^*(x_t,t)\|^2\big]} -$$

$$\min_{v'}\mathbb{E}_{x_t,t,x_0}\big[\lambda(t)\|v'(x_t,t) - \nabla_{x_t}\log q(x_t|x_0)\|^2\big]$$

Therefore, we get:

$$\mathbb{E}_{x_t,t}\big[\lambda(t)\|v(x_t,t) - v^*(x_t,t)\|^2\big] =$$

$$\mathbb{E}_{x_t,t,x_0}\big[\lambda(t)\|\nabla_{x_t}\log q(x_t|x_0)\|^2\big] - 2\mathbb{E}_{x_t,t}\big[\lambda(t)\langle v(x_t,t), v^*(x_t,t)\rangle\big] + \mathbb{E}_{x_t,t,x_0}\big[\lambda(t)\|v^*(x_t,t)\|^2\big] -$$

$$\min_{v'}\mathbb{E}_{x_t,t,x_0}\big[\lambda(t)\|v'(x_t,t) - \nabla_{x_t}\log q(x_t|x_0)\|^2\big]$$

To complete the proof, we use the relation $v(x_t,t) = \mathbb{E}_{x_0|x_t,t}\big[\nabla_{x_t}\log q(x_t|x_0)|x_t,t\big]$ from Equation 14. Integrating these components, we arrive at the final result:

$$\mathbb{E}_{x_t,t}\big[\lambda(t)\|v(x_t,t) - v^*(x_t,t)\|^2\big] =$$

$$\mathbb{E}_{x_t,t,x_0}\big[\lambda(t)\|\nabla_{x_t}\log q(x_t|x_0)\|^2\big] - 2\mathbb{E}_{x_t,t}\big[\lambda(t)\big\langle \mathbb{E}_{x_0|x_t,t}\big[\nabla_{x_t}\log q(x_t|x_0)|x_t,t\big], v^*(x_t,t)\big\rangle\big] + \mathbb{E}_{x_t,t,x_0}\big[\lambda(t)\|v^*(x_t,t)\|^2\big] -$$

$$\min_{v'}\mathbb{E}_{x_t,t,x_0}\big[\lambda(t)\|v'(x_t,t) - \nabla_{x_t}\log q(x_t|x_0)\|^2\big] =$$

$$\mathbb{E}_{x_t,t,x_0}\big[\lambda(t)\|\nabla_{x_t}\log q(x_t|x_0)\|^2\big] - 2\mathbb{E}_{x_t,t,x_0}\big[\lambda(t)\langle \nabla_{x_t}\log q(x_t|x_0), v^*(x_t,t)\rangle\big] + \mathbb{E}_{x_t,t,x_0}\big[\lambda(t)\|v^*(x_t,t)\|^2\big] -$$

$$\min_{v'}\mathbb{E}_{x_t,t,x_0}\big[\lambda(t)\|v'(x_t,t) - \nabla_{x_t}\log q(x_t|x_0)\|^2\big] =$$

$$\mathbb{E}_{x_t,t,x_0}\big[\lambda(t)\|v^*(x_t,t) - \nabla_{x_t}\log q(x_t|x_0)\|^2\big] - \min_{v'}\mathbb{E}_{x_t,t,x_0}\big[\lambda(t)\|v'(x_t,t) - \nabla_{x_t}\log q(x_t|x_0)\|^2\big].$$

$\square$

*Proof of Proposition 3.3.* Consider the problem from Proposition 3.2:

$$\min_{\theta}\Big[\mathbb{E}_{x_t,t,x_0}\big[\lambda(t)\|v^*(x_t,t) - \nabla_{x_t}\log q(x_t|x_0)\|^2\big] - \min_{\phi}\mathbb{E}_{x_t,t,x_0}\big[\lambda(t)\|v_\phi(x_t,t) - \nabla_{x_t}\log q(x_t|x_0)\|^2\big]\Big],$$

For the priors with the drift $f(t)x$ the regression target is $\nabla_{x_t} \log q(x_t|x_0) = -\frac{x_t - \alpha_t x_0}{\sigma_t^2}$. Hence one can use the parametrization $v(x_t, t) = -\frac{x_t - \alpha_t \widehat{x}_0(x_t, t)}{\sigma_t^2}$ We use reparameterization of both $v^*$ and $v_\phi$ given by:

$$v^*(x_t, t) = -\frac{x_t - \alpha_t \widehat{x}_0^*(x_t, t)}{\sigma_t^2}, \quad v_\phi(x_t, t) = -\frac{x_t - \alpha_t \widehat{x}_0^\phi(x_t, t)}{\sigma_t^2}$$

and get:

$$\min_\theta \left[ \mathbb{E}_{x_t, t, x_0} \left[ \lambda(t) \| v^*(x_t, t) - \nabla_{x_t} \log q(x_t|x_0) \|^2 \right] - \min_\phi \mathbb{E}_{x_t, t, x_0} \left[ \lambda(t) \| v_\phi(x_t, t) - \nabla_{x_t} \log q(x_t|x_0) \|^2 \right] \right] =$$

$$\min_\theta \left[ \mathbb{E}_{x_t, t, x_0} \left[ \underbrace{\lambda(t) \frac{\alpha_t^2}{\sigma_t^4}}_{\overset{\text{def}}{=} \lambda'(t)} \| \widehat{x}_0^*(x_t, t) - x_0 \|^2 \right] - \min_\phi \mathbb{E}_{x_t, t, x_0} \left[ \underbrace{\lambda(t) \frac{\alpha_t^2}{\sigma_t^4}}_{\overset{\text{def}}{=} \lambda'(t)} \| \widehat{x}_0^\phi(x_t, t) - x_0 \|^2 \right] \right] =$$

$$\min_\theta \left[ \mathbb{E}_{x_t, t, x_0} \left[ \lambda'(t) \| \widehat{x}_0^*(x_t, t) - x_0 \|^2 \right] - \min_\phi \mathbb{E}_{x_t, t, x_0} \left[ \lambda'(t) \| \widehat{x}_0^\phi(x_t, t) - x_0 \|^2 \right] \right],$$

where $\lambda'(t)$ is just another positive weighting function. $\qquad \square$

*Proof of Theorem 3.4.* In a fully analogical way, as for the unconditional case we consider the set of the Inverse Bridge Matching problems indexes by $x_T$:

$$\left\{ \min_\theta \left[ \text{KL}(\text{BM}(\Pi_{\theta|x_T}) \| M_{|x_T}^*) \right] \right\}_{x_T},$$

where $M_{|x_T}^*$ is a result of Bridge Matching conditioned on $x_T$ and $\Pi_{\theta|x_T}$ is a Mixture of Bridges for each $x_T$ constructed using bridge $q(x_t|x_0, x_T)$ and coupling $p_\theta(x_0|x_T)\delta_{x_T}(x)$.

By employing the same reasoning as in the proof of Proposition 3.1, the inverse problem can be reformulated as follows:

$$\min_\theta \mathbb{E}_{x_t, t, x_T} \left[ g^2(t) \| v(x_t, t, x_T) - v^*(x_t, t, x_T) \|^2 \right], \quad \text{s.t.}$$

$$v = \arg\min_{v'} \mathbb{E}_{x_t, t, x_0, x_T} \left[ \| v'(x_t, t, x_T) - \nabla_{x_t} \log q(x_t|x_0) \|^2 \right] dt,$$

$$(x_0, x_T) \sim p_\theta(x_0, x_T), \ t \sim U([0, T]), \ x_t \sim q(x_t|x_0, x_T).$$

Following the proof of Theorem 3.2, we obtain a tractable formulation incorporating a weighting function:

$$\min_\theta \left[ \mathbb{E}_{x_t, t, x_0, x_T} \left[ \lambda(t) \| v^*(x_t, t, x_T) - \nabla_{x_t} \log q(x_t|x_0) \|^2 \right] - \right.$$

$$\left. \min_\phi \mathbb{E}_{x_t, t, x_0, x_T} \left[ \lambda(t) \| v_\phi(x_t, t, x_T) - \nabla_{x_t} \log q(x_t|x_0) \|^2 \right] \right].$$

Utilizing the reparameterization under additional conditions (§2.3), we obtain the following representations:

$$v^*(x_t, t, x_T) = -\frac{x_t - \alpha_t \widehat{x}_0^*(x_t, t, x_T)}{\sigma_t^2}, \quad v_\phi(x_t, t, x_T) = -\frac{x_t - \alpha_t \widehat{x}_0^\phi(x_t, t, x_T)}{\sigma_t^2}.$$

Consequently, applying the proof technique from Proposition 3.3, we derive the final expression:

$$\min_\theta \left[ \mathbb{E}_{x_t, t, x_0} \left[ \lambda(t) \| \widehat{x}_0^*(x_t, t, x_T) - x_0 \|^2 \right] - \min_\phi \mathbb{E}_{x_t, t, x_0} \left[ \lambda(t) \| \widehat{x}_0^\phi(x_t, t, x_T) - x_0 \|^2 \right] \right],$$

$$(x_0, x_T) \sim p_\theta(x_0, x_T), \ t \sim U([0, T]), \ x_t \sim q(x_t|x_0, x_T).$$

$\qquad \square$

| Task | Dataset | Teacher | NFE | $L/K$ ratio | LR | Grad Updates | Noise |
|---|---|---|---|---|---|---|---|
| $4\times$ super-resolution (bicubic) | ImageNet | $I^2$SB | 1 | 5:1 | 5e-5 | 3000 | ✓ |
| $4\times$ super-resolution (pool) | ImageNet | $I^2$SB | 1 | 5:1 | 5e-5 | 3000 | ✓ |
| JPEG restoration, QF = 5 | ImageNet | $I^2$SB | 1 | 5:1 | 5e-5 | 2000 | ✓ |
| JPEG restoration, QF = 10 | ImageNet | $I^2$SB | 1 | 5:1 | 5e-5 | 3000 | ✓ |
| Center-inpainting ($128 \times 128$) | ImageNet | $I^2$SB | 4 | 5:1 | 5e-5 | 2000 | ✗ |
| Sketch to Image | Edges → Handbags | DDBM | 2 | 5:1 | 1e-5 | 300 | ✓ |
| Sketch to Image | Edges → Handbags | DDBM | 1 | 5:1 | 1e-5 | 14000 | ✓ |
| Normal to Image | DIODE-Outdoor | DDBM | 2 | 5:1 | 1e-5 | 500 | ✓ |
| Normal to Image | DIODE-Outdoor | DDBM | 1 | 5:1 | 1e-5 | 3700 | ✓ |
| Center-inpainting ($128 \times 128$) | ImageNet | DDBM | 4 | 1:1 | 3e-6 | 3000 | ✓ |

*Table 7.* Table entries specify experimental configurations: *NFE* indicates multi-step training (Sec. §3.5); *L/K* represents bridge/student gradient iteration ratios (Alg. §3.4); *Grad Updates* shows student gradient steps; *Noise* notes stochastic pipeline incorporation.

## B. Experimental details

All hyperparameters are listed in Table 7. We used batch size 256 and ema decay 0.99 for setups. For each setup, we started the student and bridge networks using checkpoints from the teacher models. In setups where the model adapts to noise: (1) We added extra layers for noise inputs (set to zero initially), (2) Noise was concatenated with input data before input it to the network. Datasets, code sources, and licenses are included in Table 8.

**Training time.** We present the training time of each in Table 9. About 75% of this training time is used to get the last 10-20% decrease of FID (e.g., drop from 3.6 to 2.5 FID in pooling SR setup or from 4.3 to 3.8 FID in JPEG with), while training for the first 25% of time already provides a good-quality model. On Sketch-to-image and Normal-to-image in multistep regime with 2 NFEs, convergence appears faster than in the corresponding single-step version.

*Table 8.* The used datasets, codes and their licenses.

| Name | URL | Citation | License |
|---|---|---|---|
| Edges→Handbags | GitHub Link | (Isola et al., 2017) | BSD |
| DIODE-Outdoor | Dataset Link | (Vasiljevic et al., 2019) | MIT |
| ImageNet | Website Link | (Deng et al., 2009) | \ |
| Guided-Diffusion | GitHub Link | (Dhariwal & Nichol, 2021) | MIT |
| $I^2$SB | GitHub Link | (Liu et al., 2023a) | CC-BY-NC-SA-4.0 |
| DDBM | GitHub Link | (Zhou et al., 2023) | \ |
| DBIM | GitHub Link | (Zheng et al., 2024) | \ |

| Task | Teacher | Dataset | Approximate time on $8\times$A100 | NFE |
|---|---|---|---|---|
| $4\times$ super-resolution (bicubic) | I2SB | Imagenet | 40 hours | 1 |
| $4\times$ super-resolution (pool) | I2SB | Imagenet | 40 hours | 1 |
| JPEG restoration, QF = 5 | I2SB | Imagenet | 40 hours | 1 |
| JPEG restoration, QF = 10 | I2SB | Imagenet | 40 hours | 1 |
| Center-inpainting ($128\times128$) | I2SB | Imagenet | 24 hours | 4 |
| Center-inpainting ($128\times128$) | DDBM | Imagenet | 12 hours | 4 |
| Sketch to Image | DDBM | Edges/Handbags | 40 hours | 1 |
| Sketch to Image | DDBM | Edges/Handbags | 1 hour | 2 |
| Normal to Image | DDBM | DIODE-Outdoor | 48 hours | 1 |
| Normal to Image | DDBM | DIODE-Outdoor | 7 hours | 2 |

*Table 9.* Training times and NFE across different tasks, teachers, and datasets.

### B.1. Distillation of $I^2$SB models.

We extended the $I^2$SB repository (see Table 8), integrating our distillation framework. The following sections outline the setups, adapted following the $I^2$SB.

**Multi-step implementation** In this setup, we use the student model's full inference process during multi-step training (Section 3.5). This means that $x_0$ is generated with inferenced of the model $G_\theta$ through *all* timesteps ($T = t_N, \ldots, t_1 = 0$) in the multi-step sequence. The generated $x_0$ is subsequently utilized in the computation of the bridge $\widehat{\mathcal{L}}_\phi$ or student $\widehat{\mathcal{L}}_\theta$ objective functions, as formalized in Algorithm 1.

$4\times$ **super-resolution.** Our implementation of the degradation operators aligns with the filters implementation proposed in DDRM (Kawar et al., 2022). Firstly, we synthesize images at $64 \times 64$ resolution, then upsample them to $256 \times 256$ to ensure dimensional consistency between clean and degraded inputs. For evaluation, we follow established benchmarks (Saharia et al., 2022; Song et al., 2023) by computing the FID on reconstructions from the full ImageNet validation set, with comparisons drawn against the training set statistics.

**JPEG restoration.** Our JPEG degradation implementation, employing two distinct quality factors (QF=5, QF=10), follows (Kawar et al., 2022). FID is evaluated on a $10,000$-image ImageNet validation subset against the full validation set's statistics, following baselines (Saharia et al., 2022; Song et al., 2023).

**Inpainting.** For the image inpainting task on ImageNet at $256 \times 256$ resolution, we utilize a fixed $128 \times 128$ centrally positioned mask, aligning with the methodologies of DBIM (Zheng et al., 2024) and CDBM (He et al., 2024). During training, the model is trained only on the masked regions, while during generation, the unmasked areas are deterministically retained from the initial corrupted image $x_T$ to preserve structural fidelity of unmasked part of images. We trained the model with 4 NFEs via the multi-step method (Section 3.5) and tested it with $1, 2$, and 4 NFEs.

### B.2. Distillation of DDBM models.

We extended the DDBM repository (Table 8) by integrating our distillation framework. Subsequent sections outline the experimental setups, adapted from the DDBM (Zheng et al., 2024).

**Multi-step implementation** In this setup, the multi-step training (Section 3.5) adopts the methodology of DMD (Yin et al., 2024a), wherein a timestep $t$ is uniformly sampled from the predefined sequence $(t_1, \ldots, t_N)$. The model $G_\theta$ then generates $x_0$ by iteratively reversing the process from the terminal timestep $t_N = T$ to the sampled intermediate timestep $t$. This generated $x_0$ is subsequently used to compute the bridge network's loss $\widehat{\mathcal{L}}_\phi$ or the student network's loss $\widehat{\mathcal{L}}_\theta$, as detailed in Algorithm 1.

**Edges $\rightarrow$ Handbags** The model was trained utilizing the Edges$\rightarrow$Handbags image-to-image translation task (Isola et al., 2017), with the $64 \times 64$ resolution images. Two versions were trained under the multi-step regime (Section 3.5), with 2 and 1 NFEs during training. Both models were evaluated using the same NFE to match training settings.

**DIODE-Outdoor** Following prior work (Zhou et al., 2023; Zheng et al., 2024; He et al., 2024), we used the DIODE outdoor dataset, preprocessed via the DBIM repository's script for training/test sets (Table 8). Two versions were trained under the multi-step regime (Section 3.5), with 2 and 1 NFEs during training. Both models were evaluated using the same NFE to match training settings.

**Inpainting** All setups matched those in Section B.1 inpainting, except we use a CBDM checkpoint (Zheng et al., 2024). This checkpoint is adjusted by the authors to: (1) condition on $x_T$ and (2) ImageNet class labels as input to guide the model. Also this is the same checkpoint used in both CDBM (He et al., 2024) and DBIM (Zheng et al., 2024) works.

## C. Additional results

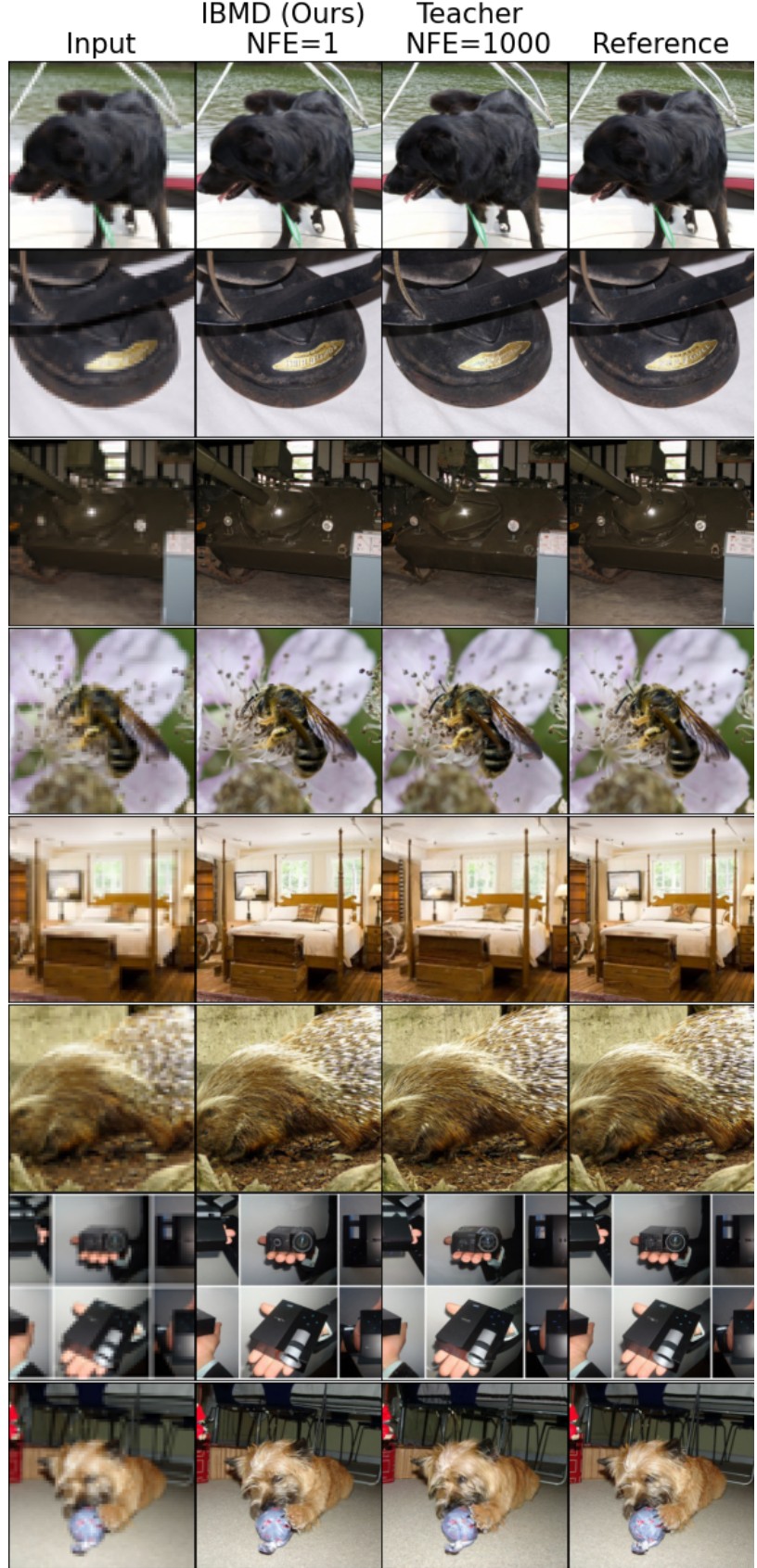

*Figure 4.* Uncurated samples for IBMD-I$^2$SB distillation of 4x-super-resolution with bicubic kernel on ImageNet $256 \times 256$ images.

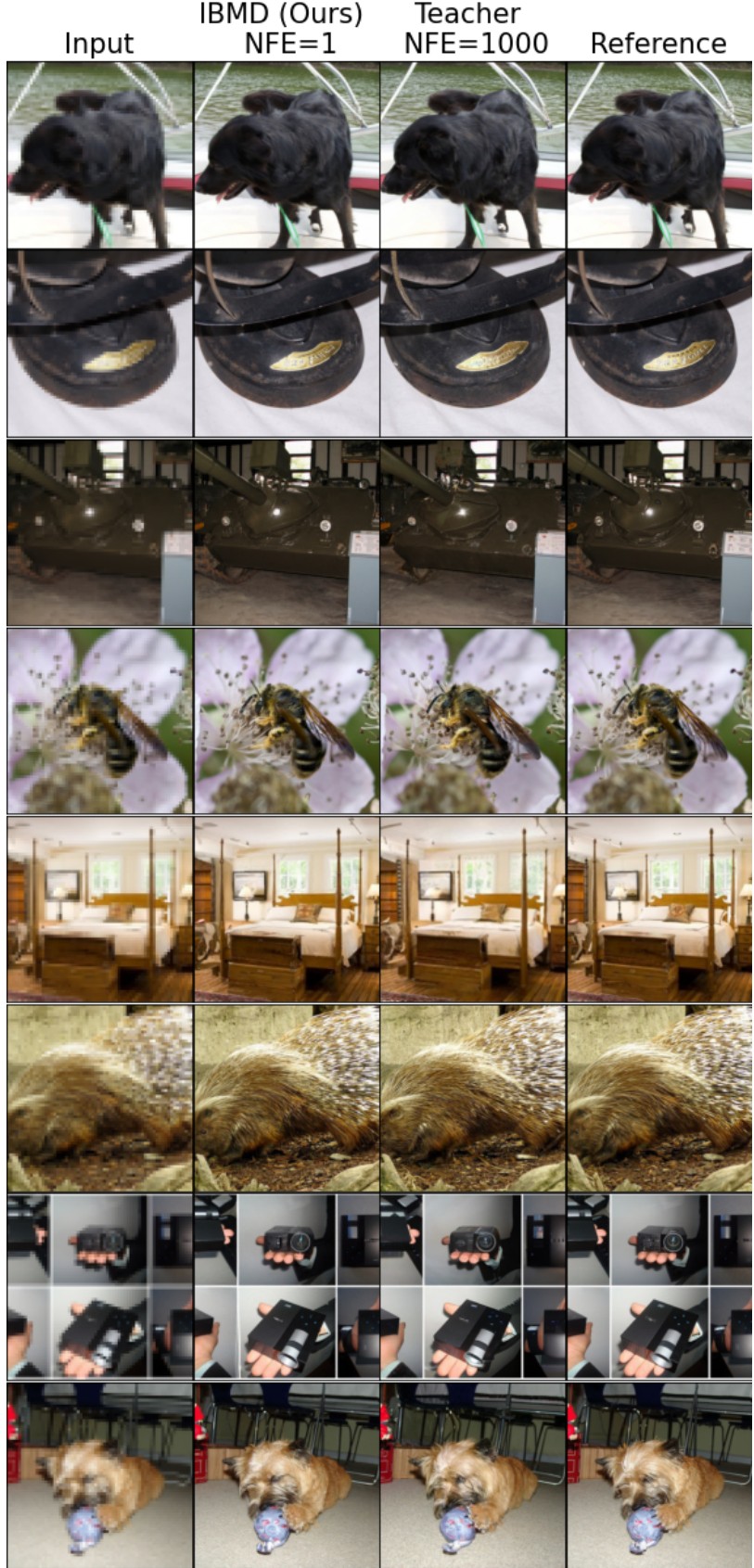

*Figure 5.* Uncurated samples for IBMD-I$^2$SB distillation of 4x-super-resolution with pool kernel on ImageNet $256 \times 256$ images.

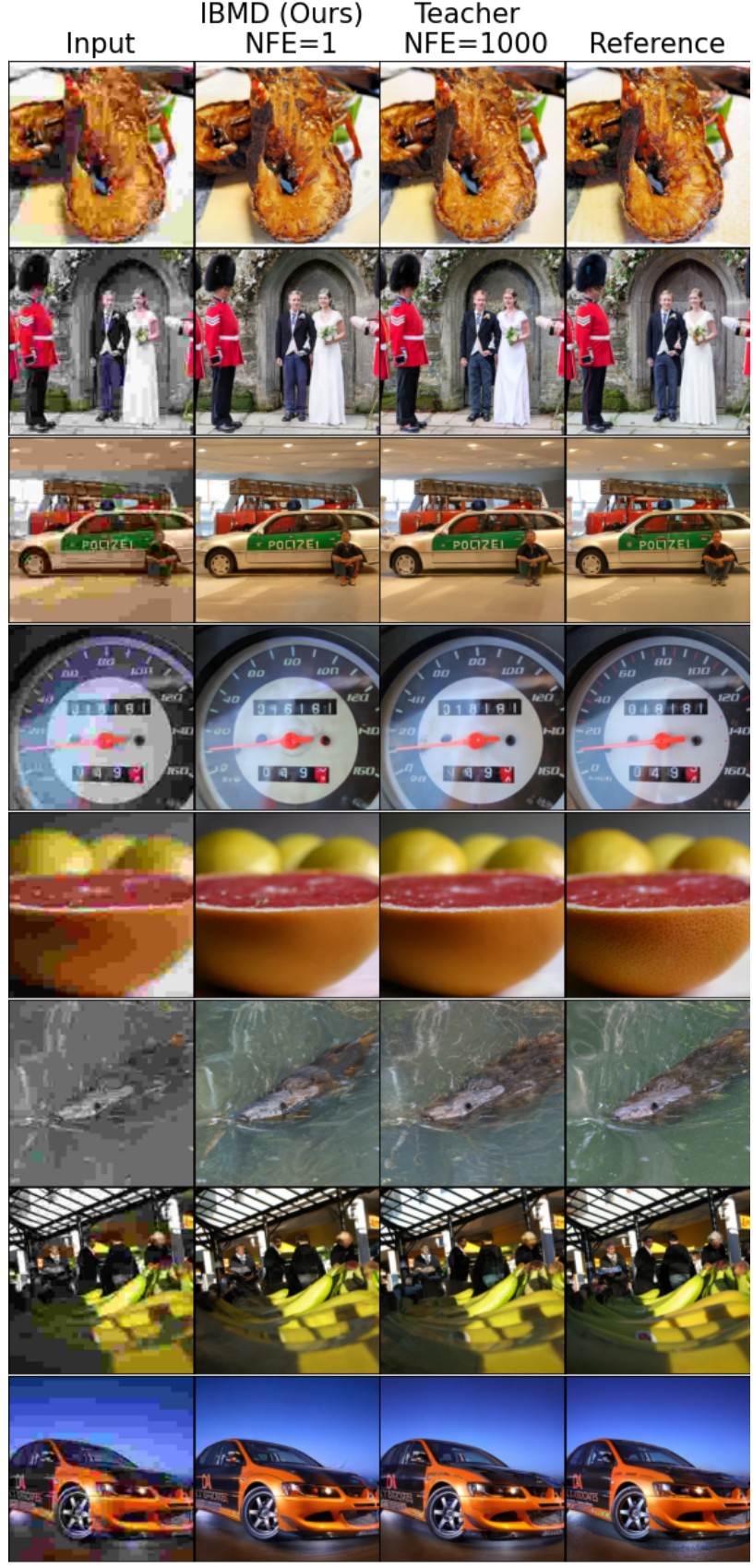

*Figure 6.* Uncurated samples for IBMD-I$^2$SB distillation of Jpeg restoration with QF=5 on ImageNet $256 \times 256$ images.

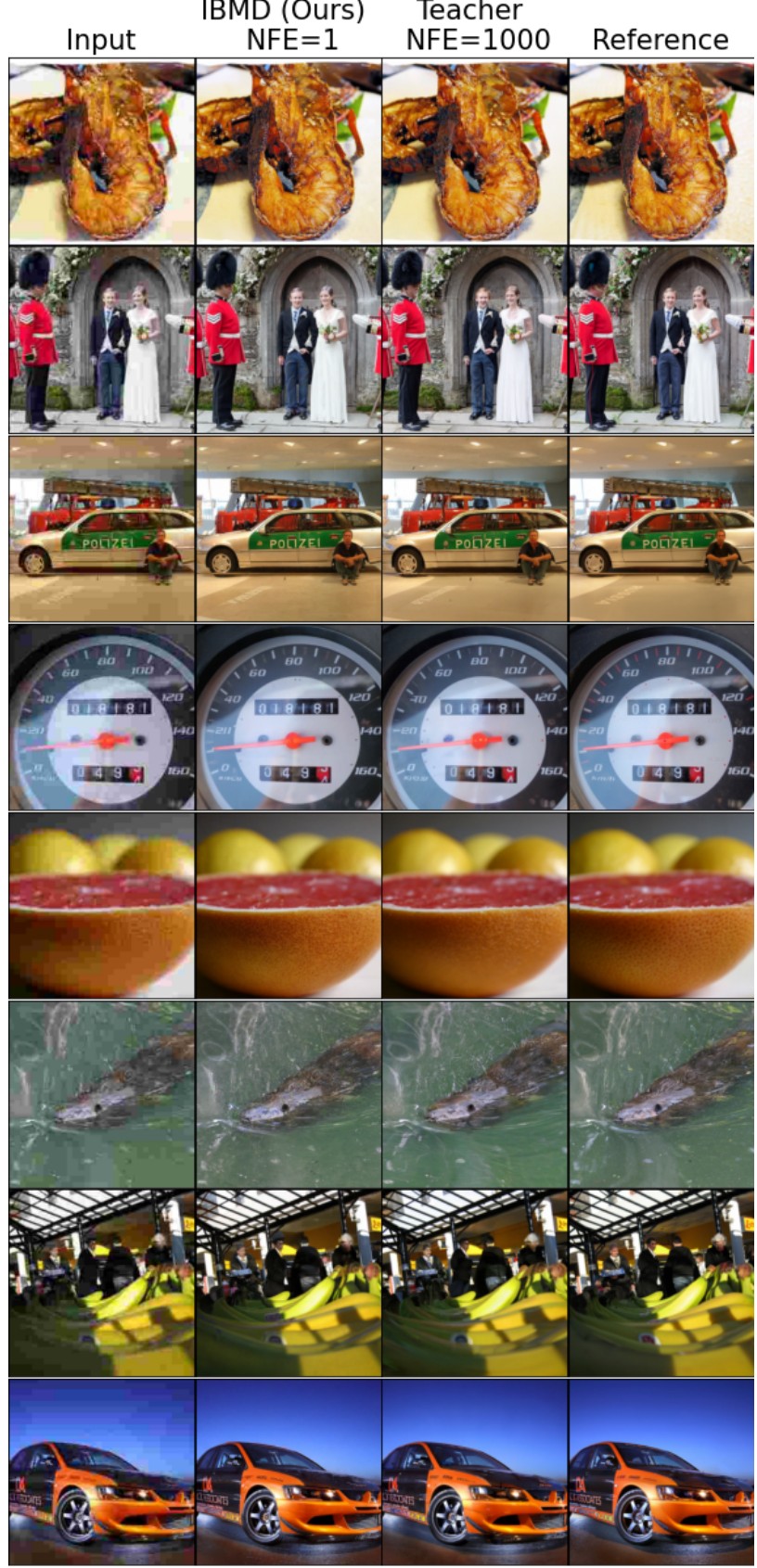

*Figure 7.* Uncurated samples for IBMD-I$^2$SB distillation of Jpeg restoration with QF=10 on ImageNet $256 \times 256$ images.

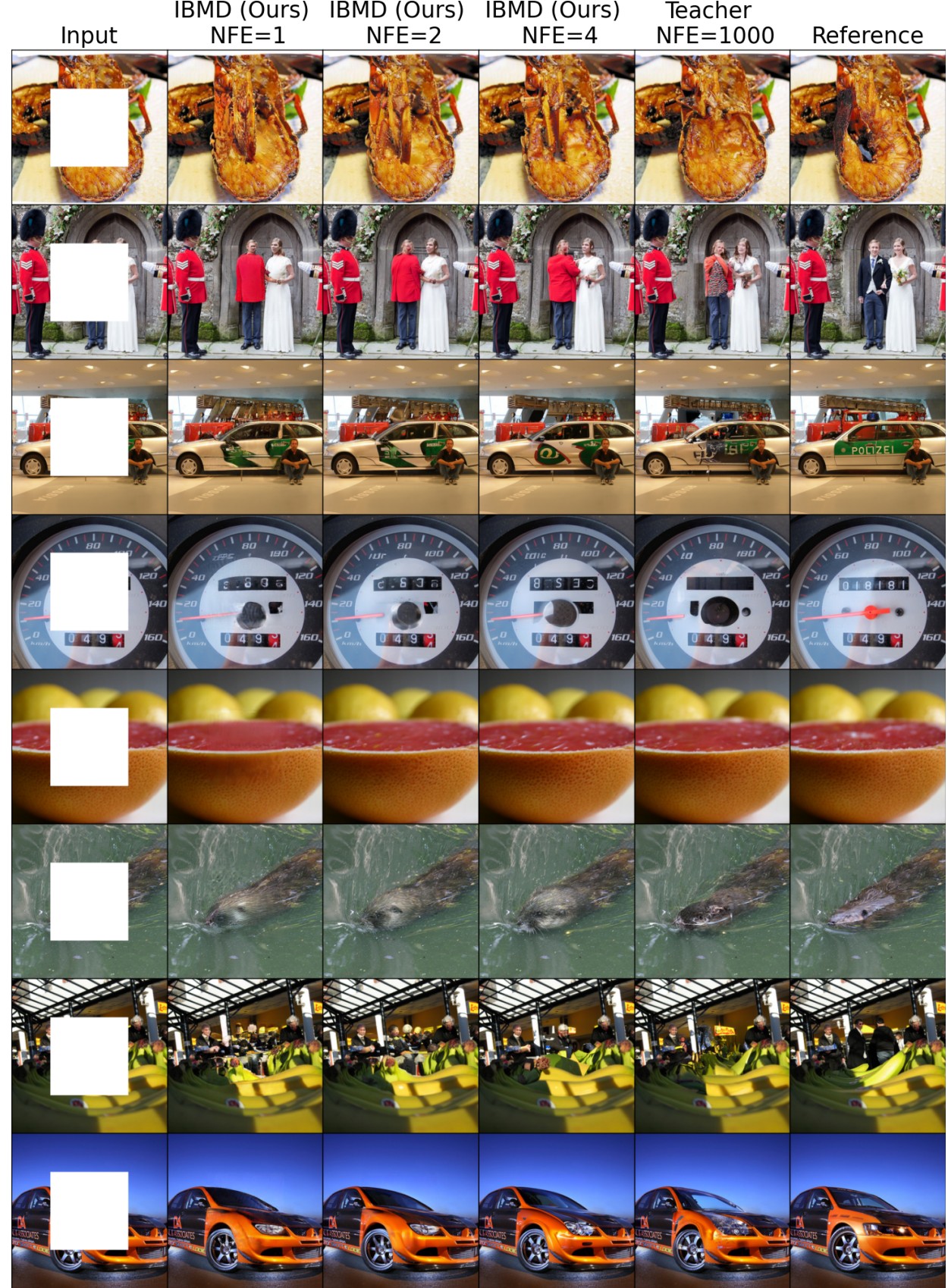

*Figure 8.* Uncurated samples for IBMD-I2SB distillation trained for inpaiting with NFE= 4 and inferenced with different inference NFE on ImageNet $256 \times 256$ images.

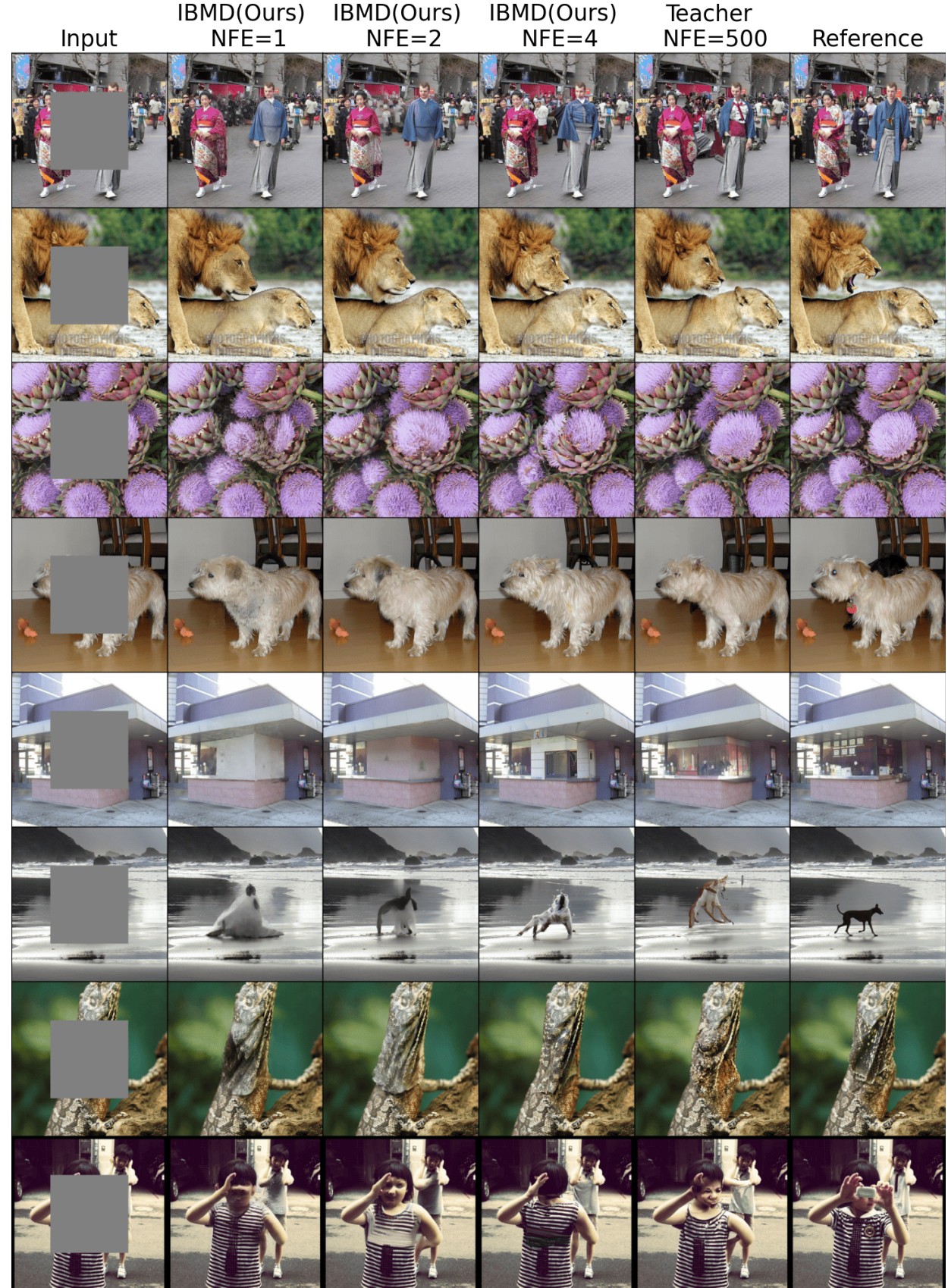

Figure 9. Uncurated samples for IBMD-DDBM distillation trained for inpaiting with NFE= 4 and inferenced with different inference NFE on ImageNet $256 \times 256$ images.

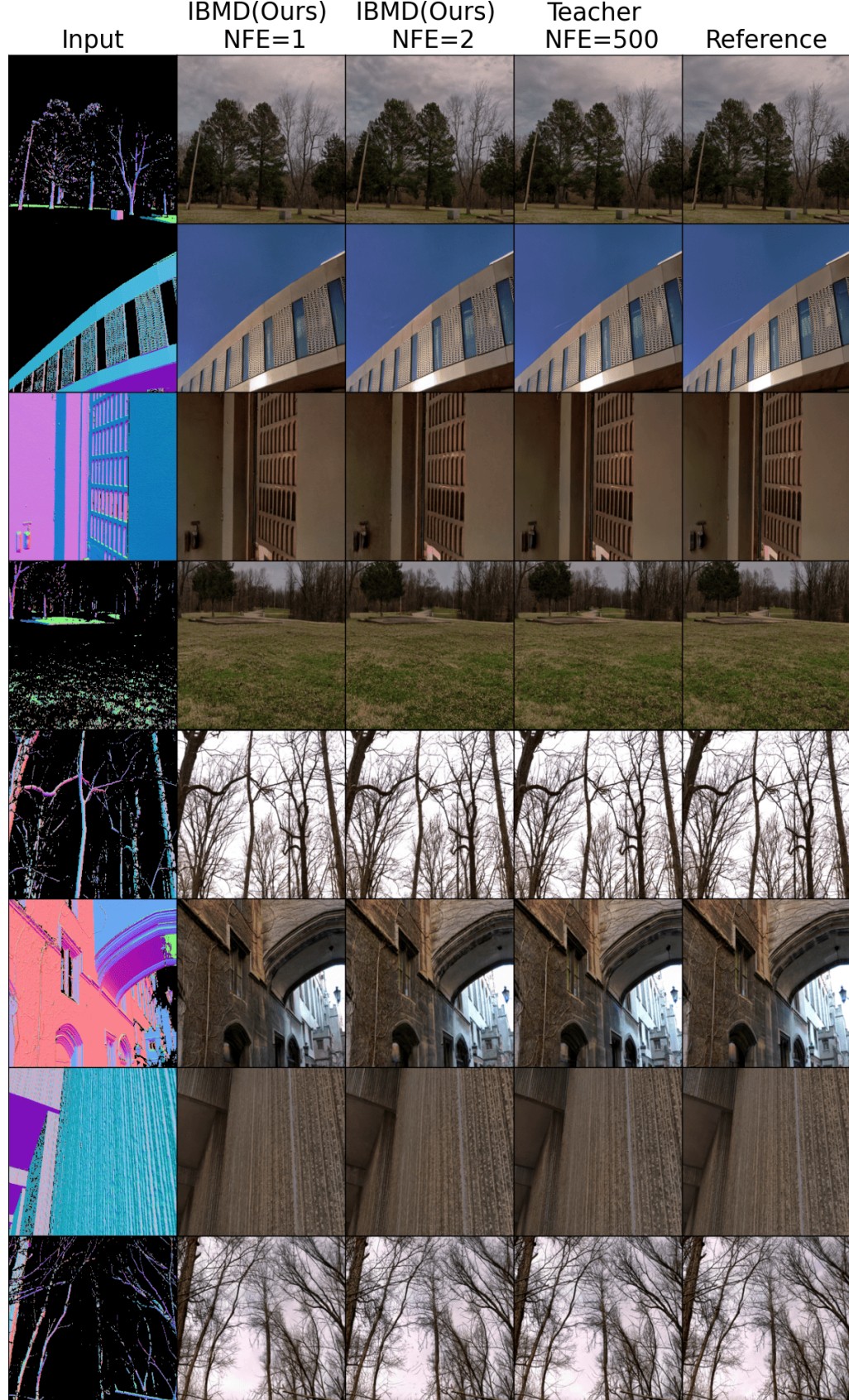

*Figure 10.* Uncurated samples from IBMD-DDBM distillation trained on the DIODE-Outdoor dataset ($256 \times 256$) with NFE= 2 and NFE= 1, inferred using the corresponding NFEs on the training set.

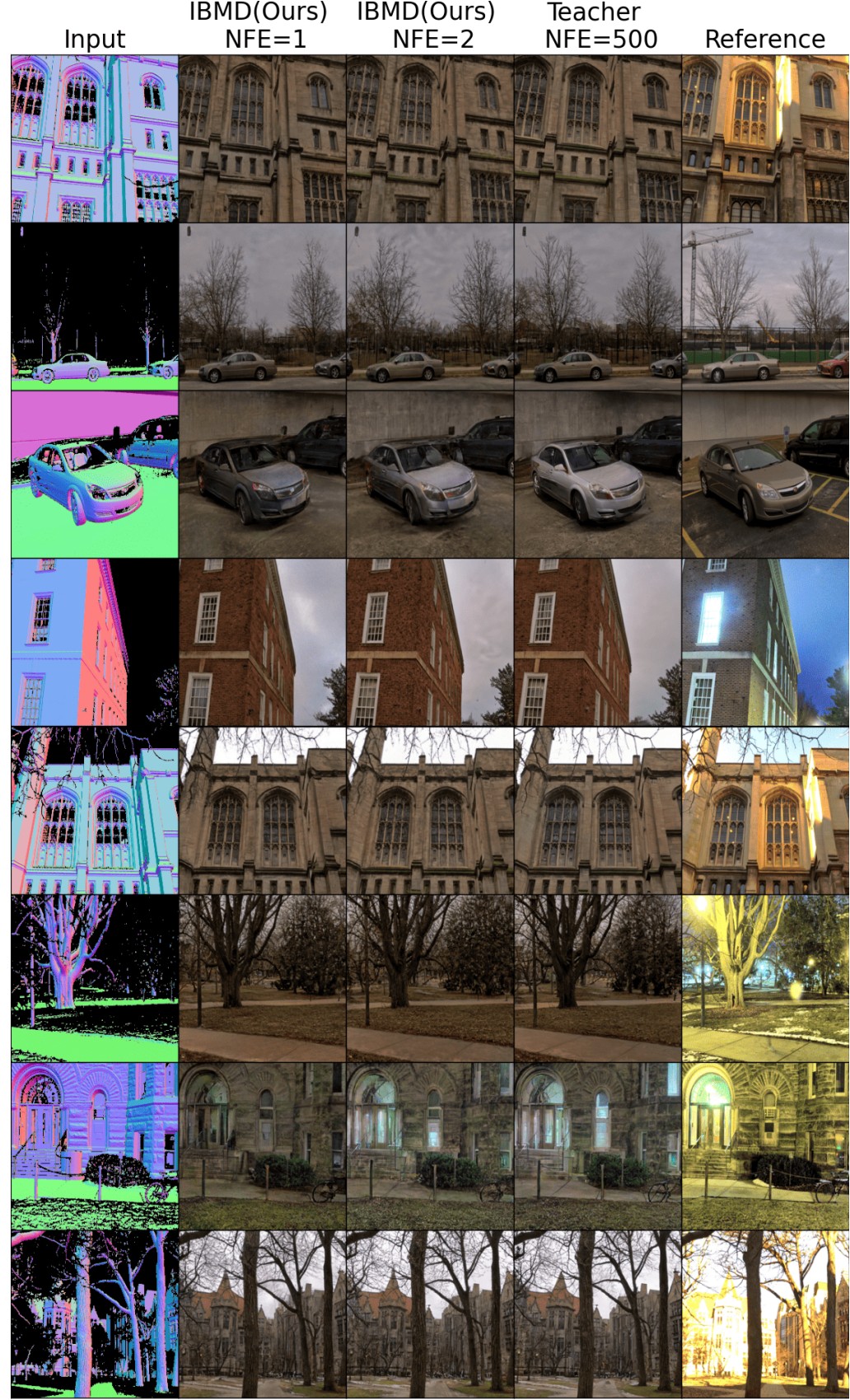

*Figure 11.* Uncurated samples from IBMD-DDBM distillation trained on the DIODE-Outdoor dataset ($256 \times 256$) with NFE= 2 and NFE= 1, inferred using the corresponding NFEs on the test set.

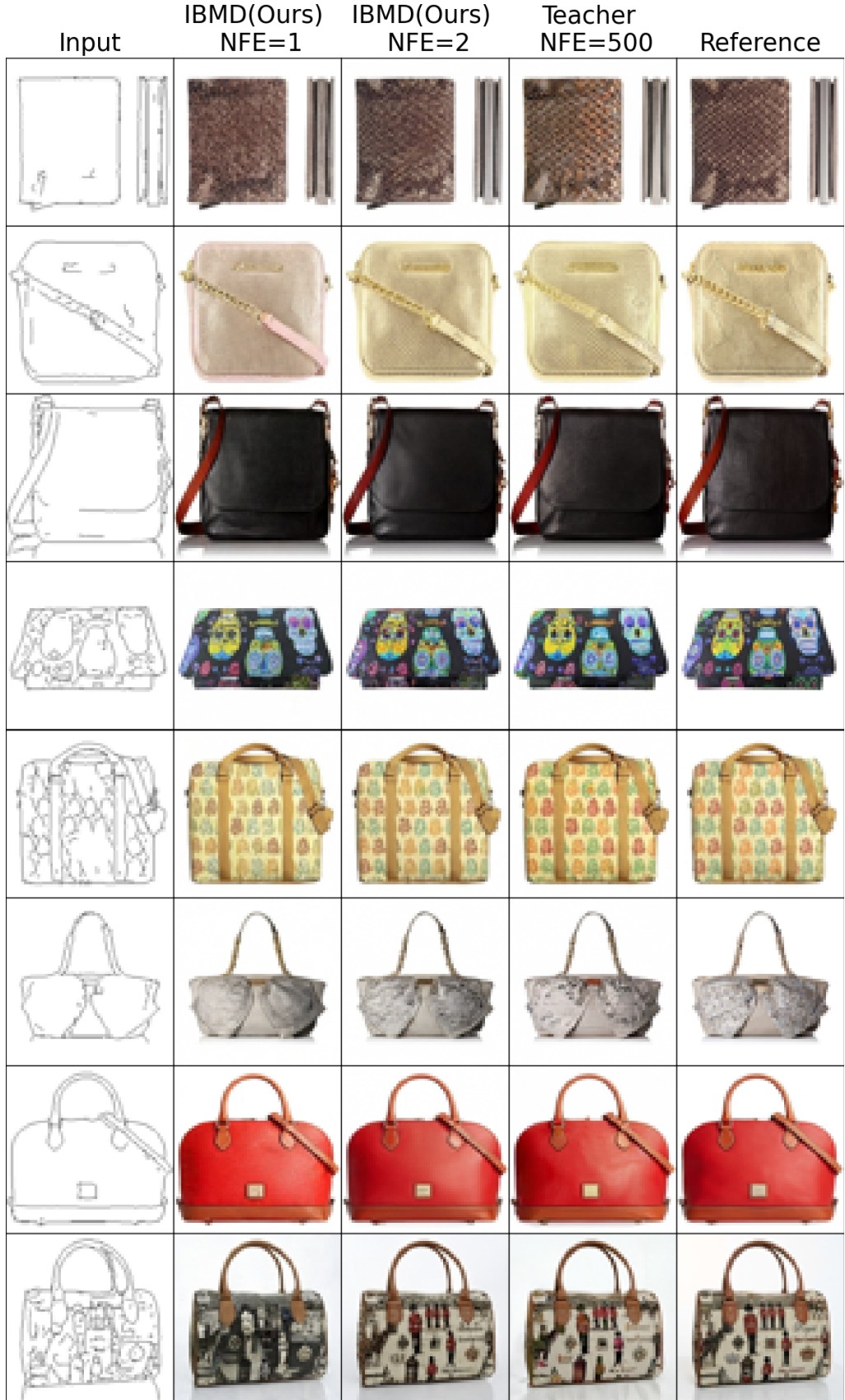

*Figure 12.* Uncurated samples from IBMD-DDBM distillation trained on the Edges → Handbags dataset (64 × 64) with NFE= 2 and NFE= 1, inferred using the corresponding NFEs on the training set.

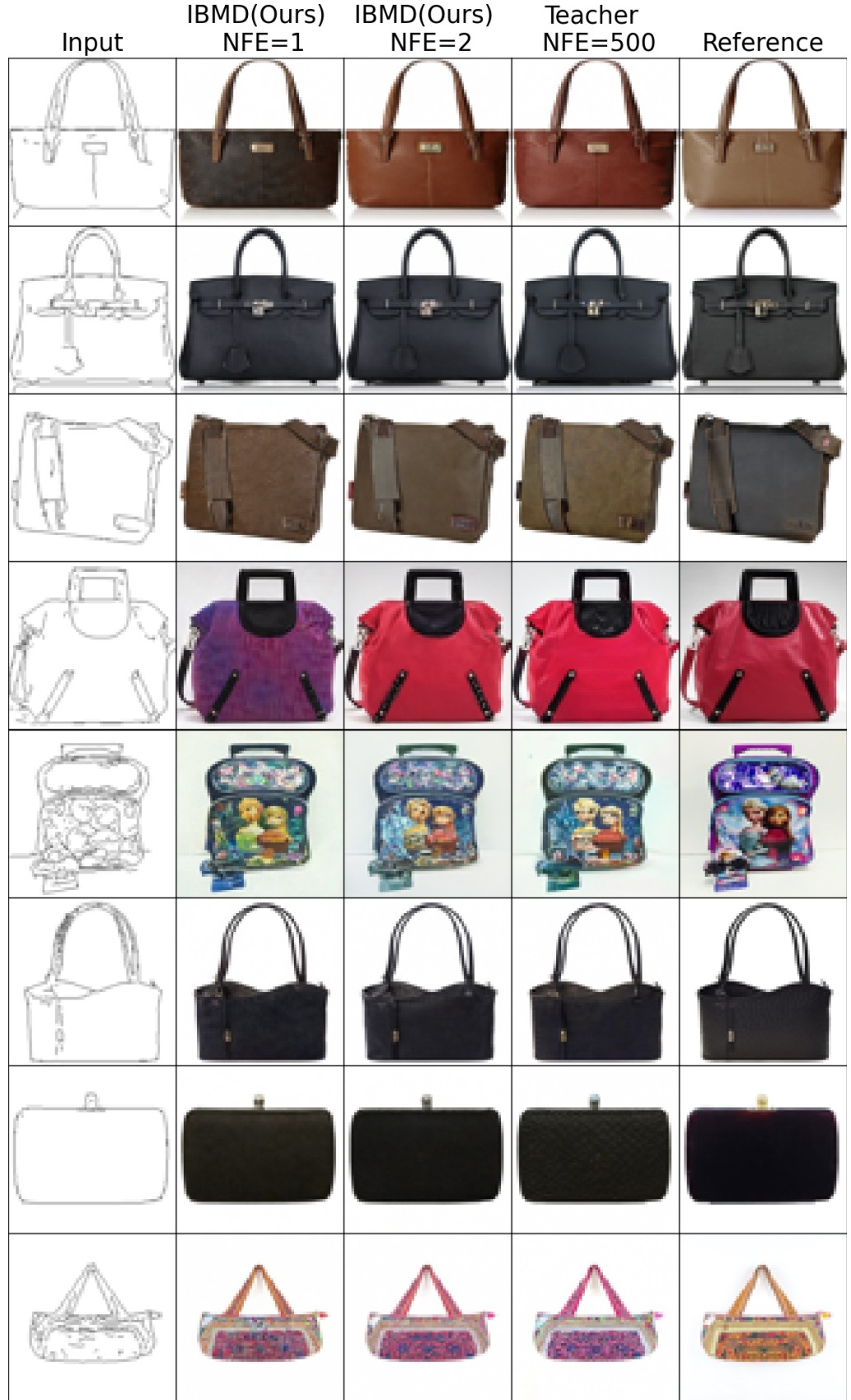

*Figure 13.* Uncurated samples from IBMD-DDBM distillation trained on the Edges → Handbags dataset (64 × 64) with NFE= 2 and NFE= 1, inferred using the corresponding NFEs on the test set.

