# OpenReview forum: "Inverse Bridge Matching Distillation"
_ICML.cc/2025/Conference — ICML 2025 poster_

### Official Review · Reviewer_kKRF · 2025-03-11

**Overall Recommendation:** 3

**Summary:**

This work incorporates the technique of score distillation in diffusion models to diffusion bridges for accelerated generation. The empirical performance demonstrates that the proposed approach is superior compared to existing baselines under multiple image-to-image tasks.

**Claims And Evidence:**

Most of the claim is well supported despite some comments on related work that needs more careful examination.

**Essential References Not Discussed:**

All key papers are cited, but they are not discussed thoroughly (which should be) in my opinion (see weaknesses).

**Experimental Designs Or Analyses:**

The experiments are well-executed.

**Methods And Evaluation Criteria:**

The proposed method and evaluation make sense for accelerating the sampling for diffusion bridges.

**Other Comments Or Suggestions:**

N/A

**Other Strengths And Weaknesses:**

## Strengths
* This work demonstrates enough empirical significance in accelerating the sampling of diffusion bridges and designing some key techniques (e.g., noise-conditioned one-step generator) tailored for distilling from diffusion bridges.

## Weaknesses
* In my opinion, although the authors mentioned some related works on distillation for diffusion models [1, 2], these works deserve a much more detailed discussion given their strong relevance — or even the use of essentially identical distillation loss functions — to the method proposed in this paper. For example, the high-level objective function of this paper is essentially the Fisher divergence used in [1], and the tractable training objective derived here is mathematically equivalent to those in [1, 2] (e.g., the loss function in this paper corresponds to the SiD ($\alpha=0.5$) loss in [1] and to the combined $\mathcal{L}_1 + \mathcal{L}_2$ loss in [2]). The derivation should be essentially the same as what has been done in [2] (same high-level objective and same final loss function). Although the focus shifts from diffusion models to diffusion bridges, and the motivation for adopting a Fisher divergence-like objective is different (which I think is a positive contribution if you can elaborate more on the connection between the KL for path measures and Fisher divergence), I still believe that better contextualization of these related works is necessary and would greatly benefit the community’s understanding of this research direction.

* I feel the claim that "previously proposed samplers and consistency models can not work with unconditional bridges" needs more justification than provided and should be re-examined. For example, [3] develops DDIM-like samplers for I2SB, which is an unconditional bridge. For consistency models, it only requires that we can simulate the PF-ODE, which I believe is plausible once we know the drift of the unconditional bridge, given it is a Markovian process.

I am happy to raise my score if these are properly addressed.

[1] Zhou, Mingyuan, et al. "Score identity distillation: Exponentially fast distillation of pretrained diffusion models for one-step generation." (ICML 2024).

[2] Huang, Zemin, et al. "Flow generator matching." arXiv:2410.19310 (2024).

[3] Wang, Yuang, et al. "Implicit Image-to-Image Schrodinger Bridge for Image Restoration." arXiv:2403.06069 (2024).

**Questions For Authors:**

N/A

**Relation To Broader Scientific Literature:**

This research is related to the broader literature on solving inverse problems.

**Theoretical Claims:**

The theoretical claims are checked in detail and are correct. However, the equivalent derivation has already been proposed in previous work, and its close connection with the proposed method in this work should at least be discussed (see weaknesses).

---

> ### Author Rebuttal · Authors · 2025-03-31
>
> Dear Reviewer kKRF, thank you for your comments.
>
> **(1) Relation to SiD [1] and FGM [2].**
> Following your suggestion, we will extend the discussion of the related work in the main text. Below, we present the preliminary version of the extension:
>
> Unlike SiD [1] and FGM [2], we focus on diffusion-bridge models used for data-to-data translation and not for generation from noise. Furthermore, our high-level objective is the KL divergence $\text{KL}(\text{BM}(\Pi_{\theta})||M^*)$ between path measures of teacher model $M^*$ and path measure $\text{BM}(\Pi_{\theta})$ given by the generator $G_{\theta}$, which differs from Fisher Divergence used in SiD [1]. The motivation of our high-level objective is to restore alignment between data pairs $(x_0, x_T)$. We derive our final tractable objective using different techniques (see Appendix A), i.e., we do not use flow product or score product identities as in [1, 2] but hypothesize that analogous identities can be derived for diffusion bridge models. Our final objective can be rewritten similarly to the final objective used in SiD [1, Eq. 23] ($\alpha=0.5$) and FGM [2, Eqs. 4.11, 4.12]:
>
> $$
> \mathbb{E}_{t, p\_{\theta}(x\_0, x\_t, x\_T)} \Big[ \underbrace{\|\widehat{x}\_0^*(x\_t, t, x\_T) - \widehat{x}\_0^{\phi}(x\_t, t, x\_T) \|^2}\_{\text{quadratic term}} + 2 \left< x_0^{*}(x_t, t, x_T) - x_0^{\phi}(x_t, t, x_T), x_0^{\phi}(x_t, t, x_T) - x_0 \right> \Big]
> $$
>
> However, in both SiD [1] (for $\alpha=1.0, 1.2$ used in experiments) and FGM [2], the authors either omitted the quadratic term or even used a negative coefficient for it in image experiments since it introduced instability. Unlike SiD [1] and FGM [2], we do not omit any parts of the original loss function and use it as the theory provides it.
>
> **Relation between KL divergence of path measures and Fisher Divergence.**
> To highlight the difference between KL divergence of path measures (which we use) and Fisher Divergence (which is used in SiD [1]), consider two reverse-time diffusions $D_1$ and $D_2$ given by the same starting distribution $p(x_T)$ and SDEs:
> $$
>     D_1: dx\_t = v(x\_t, t)dt + g^2(t)d\bar{w}\_t, \quad
>     D_2: dx\_t = \widehat{v}(x\_t, t)dt + g^2(t)d\bar{w}\_t.
> $$
> Denote marginal densities as $p(x\_t)$ for $D\_1$ and $\widehat{p}(x\_t)$ for $D\_2$, then KL divergence and Fisher Divergences are given as:
> $$
>     \text{KL}(D\_1||D\_2) = \mathbb{E}\_{t, p\_t(x\_t) } \Big[\frac{1}{2g^2(t)}\|v(x\_t, t) - \widehat{v}(x\_t, t) \|^2 \Big] + \underbrace{\text{KL}(p(x\_T)||\widehat{p}(x\_T))}\_{= 0 \text{ if } p(x\_T) = \widehat{p}(x\_T)}
> $$
> $$
>     \mathcal{L}\_{\text{SiD}}(D\_1, D\_2) = \mathbb{E}\_{t} \Big[\underbrace{\mathbb{E}\_{p(x\_t)}\|\nabla\_{x\_t}\log p(x\_t) -  \nabla\_{x\_t}\log \widehat{p}(x\_t)\|^2}\_{\text{Fisher divergence between $p(x\_t)$ and $\widehat{p}(x\_t)$}}\Big]
> $$
> Note that in SiD [1], the authors use the time average of Fisher Divergence, which compares only marginal distributions $p(x_t)$ and $\widehat{p}(x_t)$. However, two path measures with the same marginal distributions might not be equal. As a result, the minimization of Fisher Divergence with the teacher does not guarantee that the learned model will transform data in the same way as the teacher model. Nevertheless, Fisher Divergence allows one to build a Generator $x_0 = G_{\theta}(z)$ to produce data $p_{\theta}(x_0) \approx p_{\text{data}}(x_0)$, by matching marginals of data and generated samples. In contrast, we use KL-divergence between two path measures, which is zero if and only if two path measures are identical, since we need to get generations aligned to data coupling.
>
> **(2) Acceleration of the unconditional diffusion bridge models.**
>
> To obtain PF-ODE for the diffusion bridge model, one needs to subtract forward and reverse SDE drifts [4, end of Sec. 4]. For the conditional case, the drift of a forward process $p(x_0|x_T) \rightarrow \delta(x_T)$ is known analytically from Doob h-transform. In the unconditional case, the drift of a forward unconditional diffusion $p_0(x_0) \rightarrow p_T(x_T)$ is unknown. Hence, to restore PF-ODE in the unconditional case, one must learn an additional teacher model for forward-time translation, which is time-consuming.
>
> We will add that I3SB [3] is used to accelerate the unconditional bridge model. Their sampler coincides with the DBIM sampler but replaces the conditional model with an unconditional one to sample $\widehat{x}_0$. Their sampler provides good quality for moderate NFE (25+) but performs worse than distillation methods in a single NFE regime, e.g., for JPEG-10, their FID is 17, while ours is 3.8 (we will add it to the text).
>
> **Concluding remarks**.
> We would be grateful if you could let us know if our explanations have been satisfactory. If so, we kindly ask that you consider increasing your rating. We are also open to discussing any other questions you may have.
>
> References:
>
> [1,2,3] the same.
>
> [4] Shi Y. et al. Diffusion Schrödinger Bridge Matching.

---

> > ### Comment · Reviewer_kKRF · 2025-04-01
> >
> > Thank the authors for their rebuttal. Please make sure to add the discussion in the main text.
> >
> > I have a few follow-up comments:
> >
> > * I agree with your arguments about the difference between KL w.r.t. path measures and Fisher divergence, but I also think that their connection under the setting of diffusion models/diffusion bridges should be explicitly discussed. For diffusion models, the reverse drift is fully characterized by the marginal score function $\nabla_{x_t} \log p_t(x_t) = \mathbb E_{x_0 | x_t, t}[\nabla_{x_t} \log p(x_t | x_0)]$ , and thus, they are equivalent. For diffusion bridges, one can draw similar connections as the optimal drift is also given by $\mathbb E_{x_0 | x_t, t}[\nabla_{x_t} \log q(x_t | x_0)]$. The only difference I have seen here is that $\mathbb E_{x_0 | x_t, t}[\nabla_{x_t} \log q(x_t | x_0)]$ is no longer equal to the marginal score as in the case of diffusion models. Given that, the derivation of IBMD wouldn't be fundamentally different from the diffusion model case.
> >
> > * Besides, I am hesitant to accept the argument "We derive our final tractable objective using different techniques (see Appendix A), i.e., we do not use flow product or score product identities as in [1, 2]". To me, the core techniques used for the derivations in Appendix A is also the score identities used in [1, 2], i.e., $\mathbb E_{x_0, x_t} [\langle f(x_t), \nabla_{x_t} \log p_{t|0}(x_t | x_0) \rangle] = \mathbb E_{x_t} [\langle f(x_t), \mathbb E_{x_0 | x_t}[\nabla_{x_t} \log p_{t|0}(x_t | x_0)] \rangle]$.
> >
> > In conclusion, my point is that what matters here is the connection (or the technical equivalence) between them under this specific setup rather than their difference, which is valid in general but does not apply much in this setup. Although I appreciate your arguments about the differences between them, and I think this is a nice contribution to the paper.
> >
> > Nevertheless, I have adjusted my rating, and I encourage the authors to discuss the relationship between their methods and the mentioned score distillation techniques for diffusion models more thoroughly in the revised version.
> >
> > (minor)
> > * I agree with your arguments for learning the PF-ODE that characterizes the marginal distribution of the unconditional bridge is demanding. But I am wondering that, for generation purpose, one may establish the ODE w.r.t. the conditional distribution of the unconditional bridge. For example, the deterministic sampler in [3] should also correpods to an ODE trajectory?

---

> > > ### Author Response · Authors · 2025-04-04
> > >
> > > Thank you for your valuable feedback.
> > >
> > > **(1) New version of the relation.**
> > >
> > > We extended the part about the relation between KL divergence of path measures and Fisher Divergence based on your feedback. We included more details specific to the diffusion and diffusion bridge models.
> > >
> > > **Relation between KL divergence of path measures and Fisher Divergence.**
> > > To highlight the difference between KL divergence of path measures (which we use) and Fisher Divergence (which is used in SiD [1]), consider two reverse-time diffusions $D_1$ and $D_2$ given by the same starting distribution $p(x_T)$ and SDEs:
> > >
> > > $$ D\_1: dx\_t = v(x\_t, t)dt + g^2(t)d\bar{w}\_t, \quad  D\_2: dx\_t = \widehat{v}(x\_t, t)dt + g^2(t)d\bar{w}\_t.$$
> > > Let $p(x_t)$ and $\widehat{p}(x_t)$ be the corresponding marginals. Then the KL divergence and Fisher divergence are given by:
> > >
> > > $$ \text{KL}(D\_1||D\_2) = \mathbb{E}\_{t, p\_t(x\_t)} \left[\frac{1}{2g^2(t)}\|v(x\_t, t) - \widehat{v}(x\_t, t)\|^2 \right] + \underbrace{\text{KL}(p(x\_T)||\widehat{p}(x\_T))}_{= 0 \text{ if } p(x\_T) = \widehat{p}(x\_T)}, $$
> > >
> > > $$
> > >     \mathcal{L}\_{\text{SiD}}(D_1, D_2) = \mathbb{E}\_t \left[\mathbb{E}\_{p(x_t)}\|\nabla\_{x_t} \log p(x\_t) - \nabla\_{x\_t} \log \widehat{p}(x_t)\|^2 \right].
> > > $$
> > > In SiD [1], Fisher divergence is averaged over time and compares only marginal distributions $p(x_t)$ and $\widehat{p}(x_t)$ of two path measures. However, two path measures with the same marginal distributions might not be equal — thus, in general, minimizing Fisher divergence does not guarantee that $D_1 \approx D_2$ as stochastic processes. In the case of classical diffusion models where the forward drift $f(x_t, t)$ is fixed, reverse drifts are fully determined by marginal score functions:
> > > $$
> > > \widehat{v}(x_t, t) = f(x_t, t) - g^2(t)\nabla_{x_t} \log \widehat{p}(x_t), \quad v(x_t, t) = f(x_t, t) - g^2(t)\nabla_{x_t} \log p(x_t).
> > > $$
> > > Substituting these into the KL expression shows that in this specific setting — with a fixed forward SDE — KL divergence between path measures becomes equivalent (up to a constant) to the time-averaged Fisher divergence between the marginals. This explains why Fisher-based methods like SiD [1] may succeed in this context.
> > >
> > > However, this equivalence breaks down in the case of unconditional bridge matching. Here, the forward drift $f(x_t, t)$ is not fixed and depends on the data coupling $p(x_0, x_T)$. In turn, the forward drift $f_\theta(x_t, t)$ for the generated coupling $p_\theta(x_0, x_T)$ also depends on $\theta$. As a result, $f(x_t, t) \neq f_\theta(x_t, t)$, and the reverse drifts cannot be expressed solely in terms of marginal scores. Hence, KL divergence in the case of unconditional bridge matching is not equivalent to Fisher divergence between marginals. This difference is expected since, in the case of an unconditional diffusion bridge, one does not have a fixed forward process, which specifies the "dynamic part" of the measure. This highlights the importance of using KL divergence between path measures as a high-level objective instead of the previously used Fisher Divergence.
> > >
> > > **(2) Regarding score and flow product identities.** We agree that the used property is similar to score identity. We will remove this sentence in the extension of related work.
> > >
> > > **(3) PF-ODE.** In I3SB [3], the authors state in Theorem 1 that their sampler coincides with the PF-ODE of the Variance Exploding fixed Schrödinger Bridge (SB). By fixed, the authors assume that we consider SB between some distribution $p(x_0|x_T)$ and $\delta(x_T)$ for a fixed $x_T$. \textbf{This SB coincides with the forward diffusing given by Doob h-transform obtained for a VE SDE variance process, i.e., one considered in DDBM/DBIM papers.} It follows from the result that Schrödinger Bridge for VE SDE is the unique process that is Markovian and is a mixture of VE SDE bridges. Both conditions are satisfied since the authors of DDBM/DBIM use Doob h-transform for VE SDE. This result for a general Markovian SDE (not only VE SDE) can be found in [5, Theorem 2.12].
> > >
> > > The authors of I3SB in Theorem 1 omitted that $\widehat{X}_0(X_t)$ should also depend on $X_T$, i.e., one should use $\widehat{X}_0(X_t, X_T)$ since this PF-ODE is derived for the SB with a fixed $x_T$. If we use this PF-ODE for a general case of bridge diffusion from $p(x_0) \rightarrow p(x_1)$, but with $\widehat{X}_0(X_t, X_T) \approx \widehat{X}_0(X_t)$ approximated by the unconditional model, then we will indeed obtain some ODE trajectories, but we do not know any theoretical guarantees on what this ODE will produce.
> > >
> > > **Concluding remarks**.
> > > We would be grateful if you could let us know if our explanations have been satisfactory. If so, we kindly ask that you consider increasing your rating. We are also open to discussing any other questions you may have.
> > >
> > > New reference:
> > >
> > > [5] Léonard, C. (2013). A survey of the Schrödinger problem and some of its connections with optimal transport. arXiv preprint arXiv:1308.0215.

---

### Official Review · Reviewer_bxNd · 2025-03-13

**Overall Recommendation:** 4

**Summary:**

The paper introduces IBMD, an inverse bridge matching distillation method for inverse problems. The key idea is to consider bridge matching distillation as an inverse problem and convert the constrained problem into an unconstrained one using the reparameterization trick. Based on the teacher models DDBM and $I^2SB$, distilled models with IBMD achieve low FID with fewer NFEs in super-resolution, image restoration, image inpainting and image-to-image translation compared to other baselines.

## Update after rebuttal
I have no concerns regarding the submission, therefore, I will maintain my original rating of 4.

**Claims And Evidence:**

1. The proposed distillation method reduces NFEs in the inverse problem while preserving the teacher model’s performance. As far as I know, this is the first distillation approach for both unconditional and conditional inverse problems. The equations and derivations are solid.

2. It would be better to clarify the reasoning behind the statement in Section 3.2, line 240: 'The key difference in the reformulated problem is that it admits clear gradients of the generator $G_θ$.' For example, explaining that all parts are differentiable would help.

**Essential References Not Discussed:**

No.

**Experimental Designs Or Analyses:**

The experimental designs are valid.

**Methods And Evaluation Criteria:**

1. The authors follow the typical setup for inverse problems (super-resolution, image restoration, translation, and inpainting) and select appropriate teacher models (unconditional for $I^2SB$ and conditional for DDBM) along with suitable baselines.

2. One curious point is that multi-step distillation is applied to the distillation of DDBM but not to CBD and CBT. What if single-step distillation is applied to the proposed approach? How would the results change in the metric?

**Other Comments Or Suggestions:**

It would be better to use the same notation for 'single-step' and 'multi-step' in line 99.

**Other Strengths And Weaknesses:**

**Strengths**

1. Treating bridge matching distillation as an inverse problem is novel.
2. The derivation for reparameterization is solid.
3. It enables one-step inference.

**Weaknesses**

1. The qualitative results for inpainting are not satisfactory.

**Questions For Authors:**

How much time does it take to distill each model each model $I^2SB$ and DDBM?

**Relation To Broader Scientific Literature:**

Due to the reduced NFEs achieved by the proposed method, it will be more applicable to real-world inverse problem applications.

**Theoretical Claims:**

I have checked Proposition and Theorems 3.1–3.4, along with their derivations in the Appendix. The proofs seem correct.

---

> ### Author Rebuttal · Authors · 2025-03-31
>
> Dear Reviewer bxNd, thank you for your comments. Here are the answers to your questions and comments.
>
> **(1) It would be better to clarify the reasoning behind the statement in Section 3.2, line 240: 'The key difference in the reformulated problem is that it admits clear gradients of the generator .' For example, explaining that all parts are differentiable would help.**
>
> Thank you for this suggestion. In the final version, we will clarify the differentiability of all parts of the final objective.
>
> **(2) One curious point is that multi-step distillation is applied to the distillation of DDBM but not to CBD and CBT. What if single-step distillation is applied to the proposed approach? How would the results change in the metric?.**
>
> If we correctly understand, you asked about single-step distillation applied to the result of the multi-step distillation. Indeed, we applied multi-step distillation to the original teacher model, e.g., I2SB and DDBM, but not to the already distilled models like CBD or CBT. We did so, since our method is designed for distillation of diffusion-bridge models (like I2SB and DDBM), while CBD and CBT are consistency models obtained from DDBM. Result of multi-step IBMD (ours) distillation is also not a diffusion bridge model. Due to that we do not consider one-step distillation of multi-step distilled models.
>
> **(3) It would be better to use the same notation for 'single-step' and 'multi-step' in line 99.**
>
> Thank you for this suggestion. We will change it in the final version.
>
> **(4) How much time does it take to distill each model each model I2SB and DDBM?**
>
> We present the training time of each model below:
> | Task                                  | Teacher | Dataset        | Approximate time on 8A100 | NFE |
> |---------------------------------------|---------|----------------|---------------|-----|
> | $4 \times$ super-resolution (bicubic) | I2SB    | Imagenet       | 40 hours      | 1   |
> | $4 \times$ super-resolution (pool)    | I2SB    | Imagenet       | 40 hours      | 1   |
> | JPEG restoration, QF $=5$             | I2SB    | Imagenet       | 24 hours      | 1   |
> | JPEG restoration, QF $=10$            | I2SB    | Imagenet       | 40 hours      | 1   |
> | Center-inpainting ($128 × 128$)       | I2SB    | Imagenet       | 24 hours      | 4   |
> | Center-inpainting ($128 × 128$)       | DDBM    | Imagenet       | 12 hours      | 4   |
> | Sketch to Image                       | DDBM    | Edges/Handbags | 40 hours      | 1   |
> | Sketch to Image                       | DDBM    | Edges/Handbags | 1 hour        | 2   |
> | Normal to Image                       | DDBM    | DIODE-Outdoor  | 48 hours      | 1   |
> | Normal to Image                       | DDBM    | DIODE-Outdoor  | 7  hours      | 2   |
>
> About 75\% of this training time is used to get the last 10-20\% decrease of FID (e.g., drop from 3.6 to 2.5 FID in pooling SR setup or from 4.3 to 3.8 FID in JPEG with $QF=5$), while training for the first 25\% of time already provides a good-quality model. On Sketch-to-image and Normal-to-image in multistep regime with 2 NFEs, convergence appears faster than in the corresponding single-step version. We will add the approximate time used for training to Table 7 of Appendix B (Table with all hyperparameters).
>
> **Concluding remarks**.
> We would be grateful if you could let us know if the explanations we gave have been satisfactory in addressing your concerns and questions about our work. We are also open to discussing any other questions you may have.

---

> > ### Comment · Reviewer_bxNd · 2025-04-05
> >
> > Thank you for your rebuttal. I have read it carefully, along with the other reviews.
> >
> > I have no concerns regarding the submission, therefore, I will maintain my original rating of 4.

---

### Official Review · Reviewer_AN8r · 2025-03-15

**Overall Recommendation:** 4

**Summary:**

This paper proposes a new distillation scheme for diffusion bridge models. The main idea is to parameterize the entire formulation based on stochastic generator G. The student must follow input-output pairs produced by G, which constrains the path of diffusion bridge, which must coincide with the teacher path. Optimizing G, one can find a viable student. To make the formulation tractable, the constrained problem is reformulated into an unconstrained problem, resulting in a bilevel optimization problem that is somewhat GAN-like. The generator G becomes the resulting one-step generator, and the difference between the teacher error and the "student error" acts as a discriminator. G can also be designed in multi-step fashion. The entire formulation can be applied to both unconditional bridge matching and conditional bridge matching. Experiments show that the proposed method provides state-of-the-art results with much fewer steps than existing methods.

## update after rebuttal

All the reviewers have rated the paper positively. The additional results provided in the rebuttal are also convincing. I maintain my original score.

**Claims And Evidence:**

The proposed formulation based on stochastic generator G, as well as the bilevel reformulation, is sound and novel. I checked the proof and they are correct in my opinion.

**Essential References Not Discussed:**

I believe the bibliography is thorough.

**Experimental Designs Or Analyses:**

The method was evaluated on popular benchmark datasets, and the experiments show most important metrics (NFE, FID, CA). The proposed method shows state-of-the-art performance with much fewer sampling steps.

**Methods And Evaluation Criteria:**

As mentioned above, the method and the proofs are sound. The method was evaluated on popular benchmark datasets.

**Other Comments Or Suggestions:**

Please see the below question.

**Other Strengths And Weaknesses:**

The proposed distillation technique based on the G formulation is sound and novel. This particular formulation allows for the handling of both unconstrained and constrained diffusion bridge models. One downside is that the learning procedure can be quite complicated and heavy, as also pointed out in the Discussion section.

**Questions For Authors:**

The main question I have is about the meaning of the bilevel formulation. In my understanding, the second term with \phi here acts as an "expander," which means, G can become trivial (or can collapse) without this term. Initially, I thought that \phi is the student and is the resulting model. However, after reading the whole paper, I realized that G is the final model, and even if \phi looks like a student, it is actually an auxiliary component. In other words, the proposed method trains another DBM just to help train G. (This makes the whole formulation quite heavy because now we have two additional models G and \phi. How long does it take for training?)

I'd like to ask whether this interpretation is right, and I'd like to see a deeper discussion in the paper regarding the role of \phi. Currently, there is not much of a discussion about the meaning of the final formulation.

**Relation To Broader Scientific Literature:**

Diffusion bridge models are an important topic in diffusion models, and they can be used in many data-to-data translation problems. Providing a faster sampling method for diffusion bridge models can benefit many related areas.

**Theoretical Claims:**

As mentioned above, the proofs ((a) the parametrized matching problem becomes constrained optimization (9), (b) it can be reformulated as an unconstrained bilevel optimization, and (c) it can be reparametrized based on the denoisers/samplers) are sound.

---

> ### Author Rebuttal · Authors · 2025-03-31
>
> Dear Reviewer AN8r, Thank you for your comments. Here are the answers to your questions and comments.
>
> **(1)  In my understanding, the second term with \phi here acts as an "expander," which means, G can become trivial (or can collapse) without this term ... In other words, the proposed method trains another DBM just to help train $G$. I'd like to ask whether this interpretation is right, and I'd like to see a deeper discussion in the paper regarding the role of $\phi$.**
>
> Yes, this interpretation is correct. To show it more formally, note that the minimal value of the inner problem is the averaged variance of $x_0 \sim p_{\theta}(x_0|x_t, x_T)$:
>
> $$
> \min_{\phi} \mathbb{E}_{p\_{\theta}(x\_t, t, x\_0, \textcolor{MyRed}{x_T})}  \big[\lambda(t) \| \widehat{x}_0^{\phi}(x_t, t, \textcolor{MyRed}{x_T}) - x_0 \|^2 \big]\Big] = \mathbb{E}\_{p\_{\theta}(x\_t, t, x\_0, \textcolor{MyRed}{x\_T})}  \big[\lambda(t) \| \mathbb{E}\_{p\_{\theta}(x\_0|x\_t, x\_T)}[x_0] - x_0 \|^2 \big]\Big] =
> $$
> $$
> =  \mathbb{E}\_{p\_{\theta}(t, x\_t, x\_T)} \Big( \lambda(t) \Big\[\underbrace{\mathbb{E}\_{p\_{\theta}(x\_0|x\_t, x\_T)} \big[ \| \mathbb{E}\_{p\_{\theta}(x\_0|x\_t, x\_T)}[x_0] - x_0 \|^2 \big]\Big]}\_{\text{Variance of } p\_{\theta}(x\_0|x\_t, x\_T)}\Big\] \Big).
> $$
>
> For $t=T$, this is directly the variance of the generator $x_0 \sim p\_{\theta}(x_0|x_T)$. Since we are maximizing this part over $\theta$, it enforces the generator to produce more diverse outputs and avoid collapsing.
>
> Following your advice, we will add this discussion on the interpretation of how the auxiliary model $\phi$ helps to train the Generator $\theta$ to the final version of the paper.
>
> **(2) How long does it take for training?**
>
> We present the training time of each model below:
> | Task                                  | Teacher | Dataset        | Approximate time on 8A100 | NFE |
> |---------------------------------------|---------|----------------|---------------|-----|
> | $4 \times$ super-resolution (bicubic) | I2SB    | Imagenet       | 40 hours      | 1   |
> | $4 \times$ super-resolution (pool)    | I2SB    | Imagenet       | 40 hours      | 1   |
> | JPEG restoration, QF $=5$             | I2SB    | Imagenet       | 24 hours      | 1   |
> | JPEG restoration, QF $=10$            | I2SB    | Imagenet       | 40 hours      | 1   |
> | Center-inpainting ($128 × 128$)       | I2SB    | Imagenet       | 24 hours      | 4   |
> | Center-inpainting ($128 × 128$)       | DDBM    | Imagenet       | 12 hours      | 4   |
> | Sketch to Image                       | DDBM    | Edges/Handbags | 40 hours      | 1   |
> | Sketch to Image                       | DDBM    | Edges/Handbags | 1 hour        | 2   |
> | Normal to Image                       | DDBM    | DIODE-Outdoor  | 48 hours      | 1   |
> | Normal to Image                       | DDBM    | DIODE-Outdoor  | 7  hours      | 2   |
>
> About 75\% of this training time is used to get the last 10-20\% decrease of FID (e.g., drop from 3.6 to 2.5 FID in pooling SR setup or from 4.3 to 3.8 FID in JPEG with $QF=5$), while training for the first 25\% of time already provides a good-quality model. On Sketch-to-image and Normal-to-image in multistep regime with 2 NFEs, convergence appears faster than in the corresponding single-step version. We will add the approximate time used for training to Table 7 of Appendix B (Table with all hyperparameters).
>
> **Concluding remarks**.
> We would be grateful if you could let us know if the explanations we gave have been satisfactory in addressing your concerns and questions about our work. We are also open to discussing any other questions you may have.

---

### Decision · Program_Chairs · 2025-05-01

**Decision:**

Accept (poster)

**Comment:**

This paper focuses on improving the inference efficiency of diffusion bridge models, enabling single-step generation for both conditional and unconditional tasks. All reviewers recommend acceptance of this work, highlighting the novelty of the proposed method and its evaluation across a wide range of experimental settings. AC recommends acceptance. The authors are encouraged to include the reviewer discussions in the final paper.